# Genipin prevents alpha-synuclein aggregation and toxicity by affecting endocytosis, metabolism and lipid storage

Rita Rosado-Ramos[1,2,3,13], Gonçalo M. Poças [2,13], Daniela Marques[3], Alexandre Foito[4], David M. Sevillano[5], Mafalda Lopes-da-Silva[3], Luís G. Gonçalves[2], Regina Menezes[1,3,6], Marcel Ottens[5], Derek Stewart[4], Alain Ibáñez de Opakua[7], Markus Zweckstetter [7,8], Miguel C. Seabra[3], César S. Mendes[3], Tiago Fleming Outeiro[7,9,10,11,12], Pedro M. Domingos [2] & Cláudia N. Santos [1,2,3]

Parkinson's Disease (PD) is a common neurodegenerative disorder affecting millions of people worldwide for which there are only symptomatic therapies. Small molecules able to target key pathological processes in PD have emerged as interesting options for modifying disease progression. We have previously shown that a (poly)phenol-enriched fraction (PEF) of *Corema album* L. leaf extract modulates central events in PD pathogenesis, namely α-synuclein (αSyn) toxicity, aggregation and clearance. PEF was now subjected to a bio-guided fractionation with the aim of identifying the critical bioactive compound. We identified genipin, an iridoid, which relieves αSyn toxicity and aggregation. Furthermore, genipin promotes metabolic alterations and modulates lipid storage and endocytosis. Importantly, genipin was able to prevent the motor deficits caused by the overexpression of αSyn in a *Drosophila melanogaster* model of PD. These findings widens the possibility for the exploitation of genipin for PD therapeutics.

Parkinson's Disease (PD) is a common neurodegenerative movement disorder affecting millions of people worldwide. PD is a multifactorial disease and, therefore, has multiple mechanisms contributing to its pathogenesis. Huge efforts have been made both to understand the molecular basis of PD and to tackle disease progression. So far, there is no cure and the lack of effective therapeutic strategies has given PD public health priority status. In the brain, PD is characterized by the loss of dopaminergic neurons at the *substantia nigra pars compacta*. At the molecular level, it is characterized by the deposition of proteinaceous inclusions called Lewy Bodies (LB) in the surviving neurons that are mainly formed by misfolded and aggregated α-synuclein (αSyn). Aggregation of αSyn is one of the molecular hallmarks in PD, as

[1]iBET, Instituto de Biologia Experimental e Tecnológica, Oeiras, Portugal. [2]Instituto de Tecnologia Química e Biológica António Xavier, Universidade Nova de Lisboa (ITQB NOVA), Oeiras, Portugal. [3]iNOVA4Health, NOVA Medical School Faculdade de Ciências Médicas, NMS FCM, Universidade Nova de Lisboa, Lisboa, Portugal. [4]Environmental and Biochemical Sciences, The James Hutton Institute, DD2 5DA Dundee, Scotland. [5]Department of Biotechnology, Delft University of Technology, Delft, Netherlands. [6]CBIOS – Universidade Lusófona's Research Center for Biosciences & Health Technologies, Campo Grande 376, 1749-024 Lisboa, Portugal. [7]German Center for Neurodegenerative Diseases (DZNE), 37075 Göttingen, Germany. [8]Max Planck Institute for Multidisciplinary Sciences, Department of NMR-based Structural Biology, Am Fassberg 11, 37077 Göttingen, Germany. [9]Department of Experimental Neurodegeneration, Center for Biostructural Imaging of Neurodegeneration, University Medical Center Göttingen, Göttingen, Germany. [10]Translational and Clinical Research Institute, Faculty of Medical Sciences, Newcastle University, Newcastle NE2 4HH, UK. [11]Scientific employee with an honorary contract at German Center for Neurodegenerative Diseases (DZNE), 37075 Göttingen, Germany. [12]Max Planck Institute for Multidisciplinary Sciences, 37075 Göttingen, Germany. [13]These authors contributed equally: Rita Rosado-Ramos, Gonçalo M. Poças. ✉e-mail: claudia.nunes.santos@nms.unl.pt

well as increased oxidative stress and vesicular trafficking impairments. αSyn is a 14 kDa natively unfolded protein, highly expressed in human brains and it has been proposed to be a lipid-binding protein[1]. Despite its physiological function not being fully understood yet, αSyn is described as playing a role in vesicular storage and release of dopamine in neurons[2,3]. However, in pathological conditions, αSyn is described to exert toxic effects, not only by its aggregation process, but also by its interaction with membranes, through various mechanisms such as membrane thinning, pore formation, enhanced lipid flip-flop[4], and, importantly, nucleation effect[5].

(Poly)phenols have emerged as potent molecules targeting key pathological processes underlying neurodegeneration[6]. We have shown that a (poly)phenol-enriched fraction (PEF) of *C. album* L. leaf modulates central events in PD pathogenesis, by inhibiting αSyn toxicity and aggregation in cells, through promotion of the autophagic flux and a reduction of the oxidative stress[7]. In the present study, we carry out a bio-guided fractionation using a well-established yeast model of αSyn aggregation to identify genipin, the main compound exhibiting bioactivity against PD present in *C. album* PEF. Genipin modulates molecular mechanisms such as carbohydrate metabolism, endocytosis, and lipid storage in order to protect yeast cells against αSyn effects. Genipin also relieves dysfunctions in a *Drosophila melanogaster* model of PD. Altogether, our study will help to explore possibilities for therapeutic interventions in PD and other synucleinopathies.

## Results

### Bio-guided fractionation identifies genipin as a protective small molecule effective against αSyn-mediated growth impairment

We have previously described a PEF from *C. album* leaf extract capable of protecting cells against the deleterious effects of αSyn-GFP overexpression in cellular models of PD based on αSyn-GFP aggregation[7]. Myricetin-3-*O*-galactoside, myricetin-3-*O*-glucoside, and quercetin-3-*O*-galactoside were identified as the most abundant compounds in the PEF. Importantly, these compounds presented no protective effect against αSyn-GFP toxicity in any of the concentrations tested[7]. To identify the molecule(s) responsible for the protective effect, the original PEF was fractionated by reversed-phase liquid chromatography as described in methods and a total of 16 fractions (F1-F16) were obtained (Fig. 1a). The bioactivity of each fraction was tested using the yeast PD model. Briefly, yeast cells overexpressing αSyn-GFP display growth impairment, and a hint for protective bioactivity implies the capability to improve cellular growth, measured by the area under the curve (AUC). The fractions were tested at 30 μg GAE.mL$^{-1}$ and only F1 and F6 were found to significantly improve cellular growth, suggesting that they contain compound(s) conferring protection against αSyn-GFP toxicity (Fig. 1a).

All the fractions were analysed by LC-HRMS in positive and negative modes in full scan mode and data-dependent MS2 and the chromatographic peaks present in F1 and F6 were tentatively annotated by utilizing accurate mass (<3ppm). A total of 95 mass spectral features were putatively identified in F1 and F6 (Supplementary Fig. 1a and Supplementary Table 1a, b[8]). Notably, 10 of these mass spectral features were identified as belonging to the same class – iridoids. Iridoids are monoterpene compounds that comprise a cyclopentan[c]-pyran core structure and can be subdivided into carbocyclic iridoids and secoiridoids, the latter being obtained by oxidative cleavage of the 7,8-bond of the cyclopentane moiety of the carbocyclic iridoids[9–11]. Importantly, the main iridoid structures identified in F1 and F6 are carbocyclic iridoids and derivatives, with genipin derivatives (Fig. 1b) standing out among the carbocyclic iridoids identified. Considering the presence of iridoid derivatives in the bioactive fractions, we decided to test the commercially available aglycone of these derivatives, genipin, as a representative iridoid with the typical carbocyclic core structure in the bioactivity studies. Moreover, this selection of the

aglycone for further bioactivity analysis is also related to the fact that it is the cyclopentan[c]-pyran core structure that will survive gut metabolism before phase II reactions like sulfation[12]. The glucosides are hydrolysed in the gut, and therefore the aglycone is likely to have much more physiological relevance and importance to a future clinic application[12].

Yeast cells carrying the empty vector (Control cells) or overexpressing αSyn-GFP were incubated with 10 μM genipin, defined as a non-toxic concentration (Supplementary Fig. 2), and cell growth was monitored for 24 h (Fig. 1c). Incubation of yeast cells overexpressing αSyn-GFP with 10 μM of genipin prevented the growth deficits of this strain by significantly increasing the final biomass and the AUC, as can be observed in the calculated 95% confidence intervals (Fig. 1d). To test the specificity of genipin, we also evaluated any potential to prevent the growth impairment imposed by the overexpression of two other aggregation prone proteins, namely Fused in Sarcoma (FUS) and huntingtin (Htt) (Supplementary Fig. 4). The lack of effect of genipin in reducing the toxicity and aggregation of these proteins, as observed by the failed improvement on cell growth (Supplementary Fig. 4), reinforces that genipin is a specific bioactive molecule to counteract αSyn-GFP toxicity.

### Genipin reduces αSyn aggregation and toxicity in yeast cells

Beyond the ability of genipin to counteract an indirect effect of αSyn-GFP overexpression on yeast growth impairment, it is crucial to truly assess the potential of genipin to interfere with αSyn aggregation as a potential therapeutic route. We used the yeast model to evaluate the capacity of genipin to modulate αSyn-GFP toxicity, by using flow cytometry (FCM) with propidium iodide (PI) staining and αSyn-GFP aggregation, by using fluorescence microscopy. As described in the previous section, the concentration of genipin protecting yeast cells (10 μM) was confirmed as non-toxic (Fig. 2a). As expected, overexpression of αSyn-GFP during 6 h resulted in increased cell death, which was completely mitigated by the incubation with genipin (Fig. 2a). Upon αSyn-GFP overexpression, the formation of intracellular inclusions was observed in about 80% of the cells (Fig. 2b). Importantly, when incubated with genipin, the percentage of yeast cells displaying these inclusions decreased to 60 ± 6 % (Fig. 2c). This decrease in toxicity and number of cells with inclusions was not due to a decrease in αSyn-GFP expression levels (Fig. 2d). We then assessed the biochemical nature of the inclusions and found that, when incubated with genipin, yeast cells tend to form fewer SDS-insoluble inclusions or a greater proportion of smaller inclusions as indicated by the lighter spot (Fig. 2e). To further evaluate if there was an alteration in the size of αSyn-GFP inclusions, total protein extracts were separated using size exclusion chromatography (SEC). Indeed, the presence of high molecular weight species was decreased upon incubation with genipin, as can be observed by the decrease of western blot signals in the first eluted fractions, confirming a decrease in the number of bigger inclusions (Fig. 2f).

Misfolding and aggregation of αSyn has been strongly associated with deficits in protein clearance mechanisms[13–15]. Since genipin reduced the presence of larger αSyn-GFP inclusions, we followed the clearance of αSyn-GFP in yeast cells. A time-chase experiment was performed for 24 h with the de novo synthesis of αSyn-GFP blocked by using glucose repression on yeast *GAL1* promoter (Fig. 2g). We observed that αSyn-GFP clearance was not affected by the presence of genipin (Fig. 2g). Interestingly this is an aspect where genipin's lack of effect is different from the original *C. album* PEF that promoted αSyn-GFP clearance and increased autophagic flux suggesting that other protective compound(s), besides genipin, could also be present in PEF[7]. Also, the alterations in the size of αSyn-GFP aggregated species promoted by genipin were not caused by αSyn degradation by the proteasome, since the impairment in ubiquitin-proteasome system caused by αSyn expression in yeast is not relief by genipin (Fig. 2h).

## Genipin interacts with monomeric αSyn preventing its aggregation

To evaluate the possible direct interaction between genipin and αSyn, the kinetics of αSyn oligomerization was followed by thioflavin T (ThT) staining and by dynamic light scattering (DLS). As expected, a significant increase in ThT signal was detected for αSyn over time (Fig. 3a), confirming the formation of fibrils. Consistently, a significant increase in αSyn species with high molecular weight was also observed by DLS after 10 h of aggregation (Fig. 3b). When recombinant αSyn is incubated with genipin we observed that the formation of αSyn high molecular weight species and fibrils was dramatically decreased (Fig. 3a, b) suggesting that genipin interacts with monomeric αSyn, preventing its aggregation. Moreover, at 25 h of oligomerization, it was observed that the levels of intermediate oligomeric species were strongly decreased by genipin (Fig. 3c). After 68 h of aggregation, αSyn

species were separated by SEC and monomeric αSyn was only detected when the incubation with genipin was performed (Fig. 3d, black arrow). Using transmission electron microscopy (TEM), it was observed that vehicle-treated αSyn formed typical amyloid fibrils at 68 h, whereas genipin-treated αSyn formed only smaller aggregates (Fig. 3e).

To gain insight into the molecular basis of the interaction of genipin with αSyn, we recorded two-dimensional NMR spectra of αSyn in the absence or presence of 100-fold excess of genipin (Fig. 3f). The $^1H$-$^{15}N$ correlation spectrum, which was started after 45 minutes of incubation of αSyn with genipin, was similar to the spectrum recorded for αSyn alone (Fig. 3f, black spectrum). Thus, αSyn remained disordered despite the large excess of genipin. Genipin effect on αSyn structure is assessed by detecting changes in NMR signal intensity. A ~5−10% decrease in signal intensity was, however, observed for several

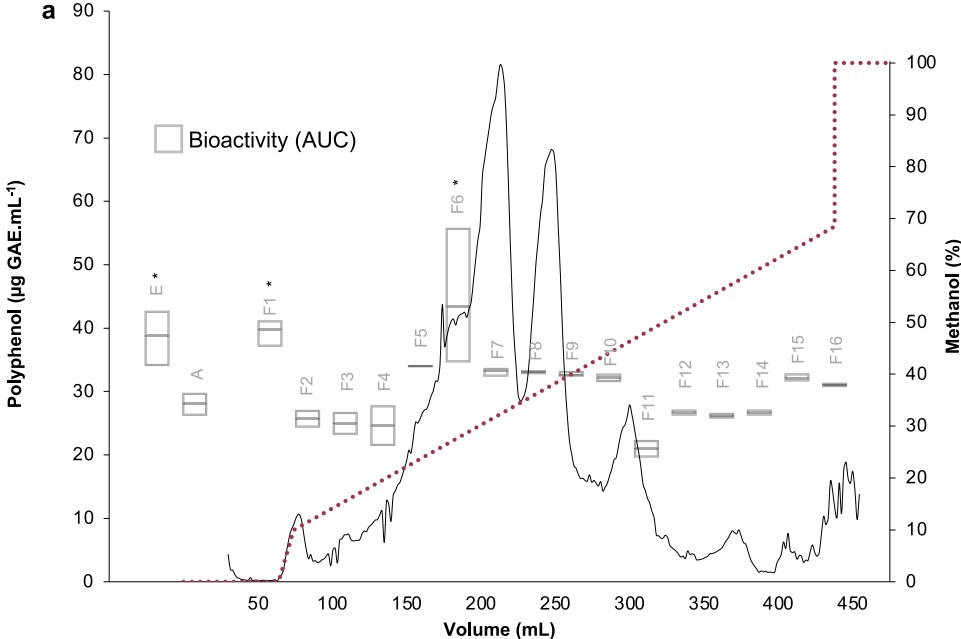

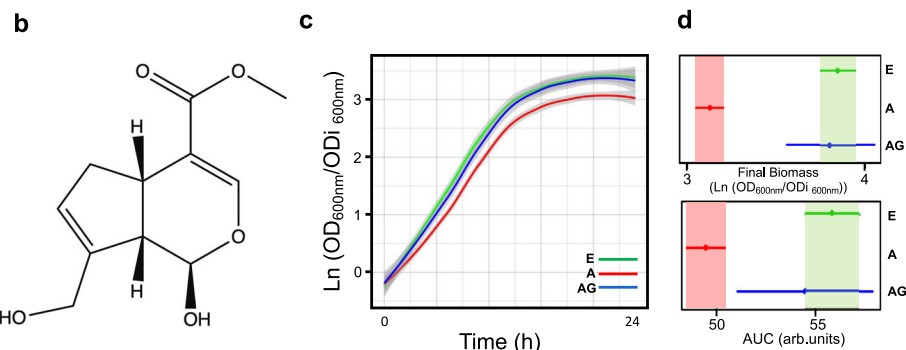

**Fig. 1 | Bioguided fractionation of *Corema album* L. leaf PEF extract identifies genipin derivatives as the bioactive compounds. a** PEF was fractionated by LC with a methanol gradient (pointed purple line, right y-axis). The chromatogram was obtained by quantification of total phenols (dashed gray line, left y-axis; µg GAE.mL⁻¹: gallic acid equivalent per mL) for the fractions of *C. album*. L. A total of 16 fractions were obtained and tested in the yeast PD model. Control cells carrying the empty vector and cells expressing two copies of αSyn-GFP were incubated in galactose medium supplemented or not with 30 µg GAE.mL⁻¹ for each fraction for 24 h. $OD_{600nm}$ was monitored every hour. The bioactivity is measured by the area

under the curve (AUC). E vs A, *p = 0.0202, F1 vs A, *p = 0.0102, F6 vs A, *p = 0.0289, (n = 3) **b** Structure of genipin, an iridoid monoterpene compound. **c** Growth curves of αSyn-GFP yeast in the presence or not of genipin. Growth curves were treated using R package grofit to adjust a nonlinear parametric regression (model-based) and the growth parameters were estimated from the best model fit. **d** Final biomass and AUC 95 % confidence intervals were used to assess statistical differences between conditions and are represented as horizontal lines. Control cells with empty vector– E, αSyn-GFP cells – A, αSyn-GFP cells +10 µM genipin – AG.

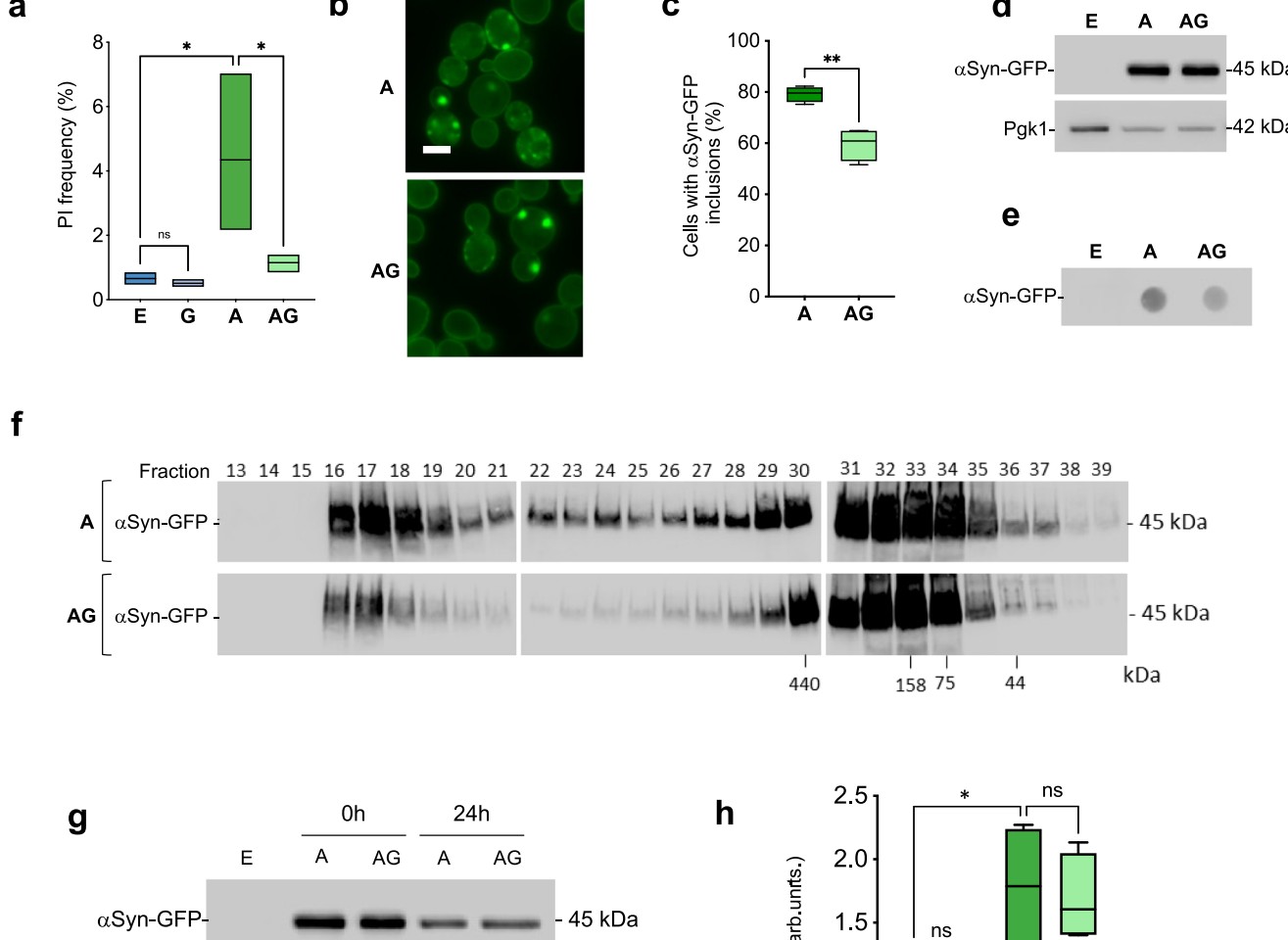

**Fig. 2 | Genipin reduces αSyn aggregation and toxicity. a** αSyn-GFP toxicity was assessed by FCM. E vs A, *$p$ = 0.0181; A vs AG, *$p$ = 0.0159 ($n$ = 3). **b** Fluorescence microscopy imaging of αSyn-GFP-overexpressing cells supplemented or not with 10 μM of genipin (scale bar 5 μm) ($n$ = 4). **c** Percentage of cells with αSyn-GFP inclusions (A, 1659 cells; AG, 2044 cells) cells for were counted. A vs AG, **$p$ = 0.0014 ($n$ = 4) **d** αSyn-GFP protein levels of cells subjected to the conditions indicated (Pgk1 was used as loading control), assessed by western blot ($n$ = 3). **e** Filter trap assay of total yeast protein extract. Quantification of αSyn signal retained in the membrane by western blot ($n$ = 3). **f** Total yeast protein extract was resolved using size exclusion-chromatography (SEC). Fractions were collected and the αSyn-GFP species present were analyzed by SDS-PAGE followed by western blot ($n$ = 3). **g** Clearance of αSyn-GFP was evaluated by western blotting at the indicated time points. Pgk1 was used as loading control ($n$ = 3). **h** Proteasome impairment evaluated by FCM using uGFP, Median Fluorescence Intensity (MFI) ($n$ = 4, E vs A, *$p$ = 0.0121 ($n$ = 4). Control cells with empty vector – E, Control cells + 10 μM genipin – G, αSyn-GFP cells – A, αSyn-GFP cells + 10 μM genipin – AG.

residues in the N-terminal half of αSyn (Fig. 3g; black bars). We then re-measured the two-dimensional $^1$H-$^{15}$N correlation spectrum after effective incubation times of 20 h (Fig. 3f, green spectrum) and 3.5 days (Fig. 3f, red spectrum). After these prolonged incubation times, signals of the N-terminal 60 residues were strongly attenuated (Fig. 3g). Indeed after 3.5 days, most cross-peaks of residues 1-64 were broadened beyond detection (Fig. 3g, red bars). In addition, the signals up to residue 105 were very weak (Fig. 3f, g). The signal intensity profile within the region from residue 64-105 was also recapitulated in the chemical shift perturbation profile (Fig. 3h). The reduced signal intensity correlates with the presence of lysines (Fig. 3g, blue dots) and can be explained by genipin-mediated cross-linking of lysines[16]. Inter-estingly, since αSyn has no arginines, all fully positively charged

residues at physiological pH are lysines, which makes αSyn a good target for genipin.

## The protective effect of genipin implicates central biological processes of PD pathology

To evaluate the potential biological targets modulated by αSyn-GFP and by genipin, we profiled the transcriptome of both control cells and αSyn-GFP overexpressing cells incubated or not with genipin using RNA-seq. Interestingly, the incubation of control cells with genipin did not result in any differential gene expression comparing to the control cells (carrying the empty vector, Fig. 4c). Overexpression of αSyn-GFP during 6 h resulted in 63 genes differentially expressed comparing to the control cells with the empty vector (Fig. 4a and

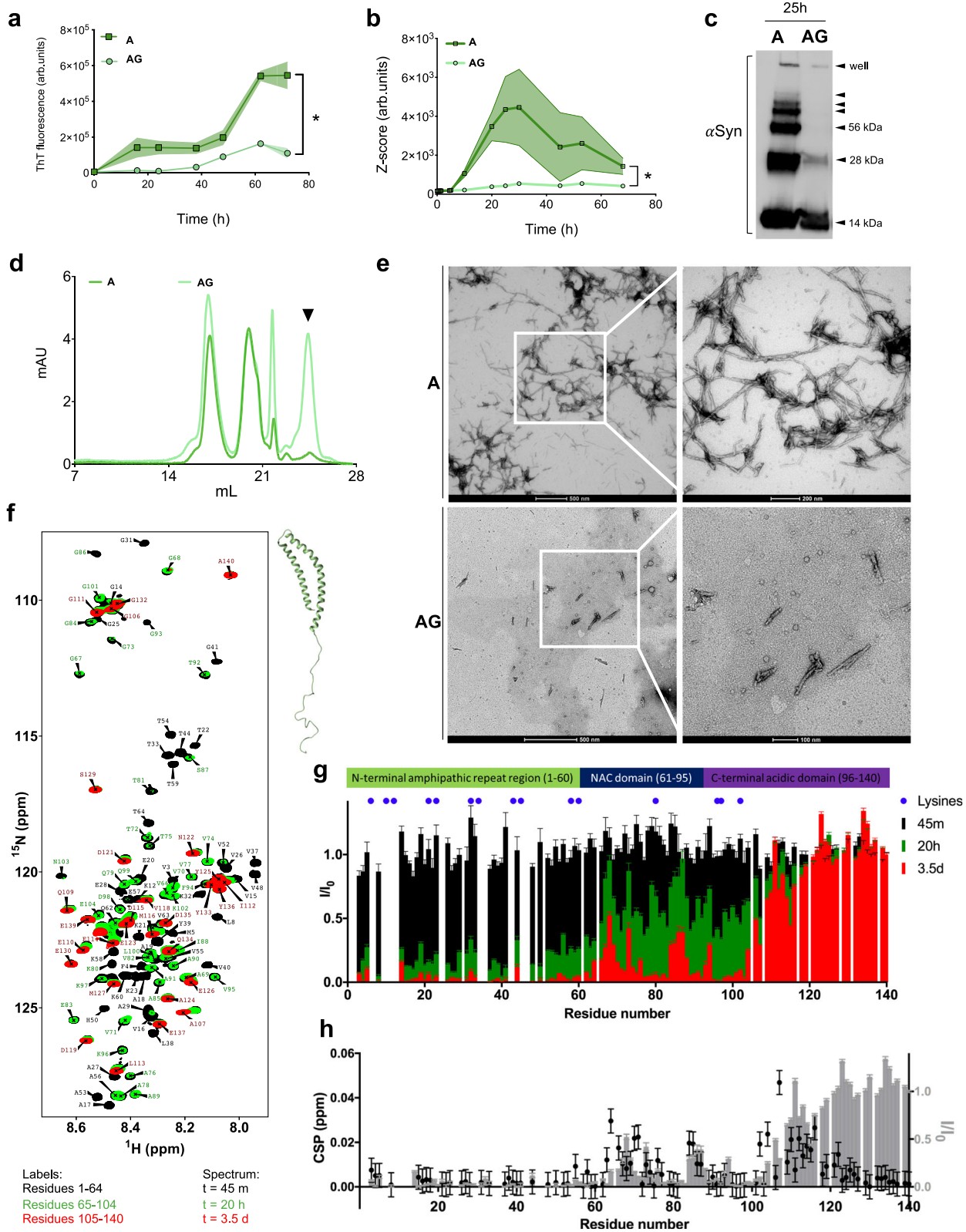

Supplementary Table 2). Impressively, the overexpression of αSyn-GFP during 6 h, treated with genipin, resulted in 437 genes differentially expressed compared to the control cells carrying the empty vector (Supplementary Table 3). Among these 437 genes, 47 are genes also affected by the overexpression of αSyn-GFP alone. These 47 genes included the same 30 upregulated and the same 17 downregulated in both conditions (highlighted in Supplementary Table 3). Among these,

18 genes that were modulated by αSyn-GFP in cells after treatment with genipin attained levels in a direction that would meet the control values, whilst 29 genes are modulated in a manner contrary to that seen in control cells with the empty vector.

The transcriptomes of αSyn-GFP overexpressing cells with/without genipin treatment were compared to uncover possible targets in the context of genipin's protection. Importantly, 235 genes were

**Fig. 3 | Genipin interacts with αSyn in vitro and hampers the formation of high molecular weight species.** Pure αSyn (70 μM) oligomerization was allowed in the presence of genipin. **a** Fibrilization kinetics of recombinant αSyn assessed by Thioflavin-T reaction. αSyn alone (dark green square) or with 10 μM of genipin (light green circle). A vs AG, *p = 0.0430 (n = 2). **b** Aggregation was monitored by Dynamic Light Scattering (DLS) over time - αSyn alone (dark green square) or with 10 μM of genipin (light green circle) (n = 2, *p = 0.0137). **c** αSyn species obtained after 25 h of in vitro oligomerization were separated by SDS-PAGE followed by western blot (n = 2). **d** Chromatogram of the size exclusion-chromatography (SEC) of the αSyn aggregated species in vitro after 68 h of incubation recorder absorbance at 280 nm (mAU). αSyn alone (dark green line) or with 10 μM of genipin (light green line) (n = 2). **e** TEM images of the obtained in vitro αSyn fibrils after 68 h of incubation. Scale bar 500, 200 nm and 100 nm as indicated (n = 2). **f** Superposition of 2D $^{1}$H-$^{15}$N HSQC NMR spectra of recombinant $^{15}$N-labelled αSyn incubated with genipin for 45 min (black), 20 h (green) and 3.5 days (red). Structure of human micelle-bound αSyn (1XQ8, PDB DOI: 10.2210/pdb1XQ8/pdb). **g** Residue-specific intensity changes in $^{1}$H-$^{15}$N HSQC signal of αSyn incubated with genipin for 45 min (black), 20 h (green) and 3.5 days (red). Error bars were estimated on the basis of the signal-to-noise in the NMR spectra. **h** Residue-specific chemical shift perturbations overlapped with $^{1}$H-$^{15}$N HSQC signal of αSyn incubated with genipin for 3.5 days. Human αSyn– A, human αSyn + 10 μM genipin – AG.

differentially expressed between these two conditions (Fig. 4b and Supplementary Table 4). By mapping the known intracellular distribution of the differentially expressed genes (Fig. 4d), we identified that genipin treatment is acting in central processes implicated in PD pathology, such as carbohydrate metabolism, degradation processes and trafficking-related pathways.

Furthermore, Gene Ontology (GO) analysis of statistically significant biological processes affected upon genipin treatment, confirmed that processes involving both carbohydrate (GO:0006000, GO:0000025, GO:0006123, GO:0046835, GO:0015755) and lipid metabolism (GO:0032369, GO:0015880) were affected, as well as protein degradation and trafficking pathways (GO:1901799, GO:0000338, GO:0032456, GO:0000425, GO:0032447, GO:0051085) (Fig. 4e). This suggests that genipin treatment could have a pleotropic effect, exerting its protective action through multiple cellular mechanisms, downstream or concomitant to its direct interaction with αSyn which needs to be further investigated.

## Genipin decreases oxidative stress and modulates carbon metabolism

The finding that carbohydrate metabolic processes are among the most affected biological processes by genipin protection (Fig. 4e) suggests a possible role of central carbon metabolism. Bioenergetics, redox homeostasis, and central carbon metabolism have been implicated in PD pathology. Also, toxins that are described to cause PD-like symptoms are known to affect cellular respiration and to cause mitochondrial dysfunction, which has been well established as a causative mediator of PD pathogenesis[17]. Moreover, metabolomic data from clinical and experimental models of PD point to energy (TCA cycle and glycolysis), lipids and fatty acid metabolic processes as the most affected in a disease scenario[18].

Our RNA-seq data indicates that genipin treatment induces the downregulation of a gene encoding for lactate transporter (*JEN1*) and the upregulation of genes encoding for glucose (*HXT10*) and glycerol (*GUP2*) transporters. Lactate levels were increased upon genipin treatment, although not significant (Fig. 5a, compound 1) while glycerol levels were found unaltered (Fig. 5a, compound 2). Moreover, an up-regulation of the alternative cytochrome b2, *CYB2*, which reduces lactate to pyruvate, was also observed (Fig. 5a, d).

Looking in more detail into glycolysis, we observed an upregulation of *PFK27* and *FBP26* genes in αSyn-GFP overexpressing cells upon genipin treatment (Fig. 5a). Citrate levels were increased upon αSyn-GFP overexpression and were not altered by genipin treatment (Fig. 5a, compound 4). Moreover, the non-mitochondrial citrate synthase (*CIT2*), was found to be up-regulated in cells overexpressing αSyn-GFP incubated with genipin, comparing with control cells (Fig. 5a). For succinate levels, although not significant, there was a trend of decreasing levels upon ααSyn-GFP overexpression, which was slightly compensated by genipin treatment (Fig. 5a, compound 5). As ornithine and glutamate can contribute to TCA cycle, their levels were also assessed. In both cases, there was an increase in their levels upon incubation of control cells with genipin, although not significant for glutamate (Fig. 5a, compound 6, and compound 7). Moreover, the

levels were not affected in αSyn-GFP overexpressing cells per se. Remarkably, in αSyn-GFP overexpressing cells, genipin sustained the increase in the levels of both metabolites. Also, acetate levels were not altered (Fig. 5a, compound 3).

Furthermore, transcriptomic analysis revealed the downregulation of genes of mitochondrial Complex IV of the electron transport chain *COX4*, *COX13*, and COX14, with *COX4* and *COX13* downregulation upon genipin treatment being validated by RT-qPCR (Figs. 5a, 5b, 5c). Noteworthy, *COX4* and *COX13*, both of which were upregulated by αSyn-GFP overexpression itself, were decreased to control levels upon simultaneous incubation of genipin and αSyn-GFP overexpression (Figs. 5b, 5c). The downregulation of complex IV genes in genipin-treated cells might indicate a modulation of oxidative phosphorylation, which is the main source of ROS, namely superoxide ($O_2^{·-}$). In fact, upon αSyn-GFP overexpression, the levels of $O_2^{·-}$ were increased as expected, while genipin treatment prevented yeast cells from exhibiting this increase (Fig. 5e).

Metabolic processes such as the response to nutrients, protein and lipid synthesis, as well as catabolic processes such as autophagy, are regulated by the crucial player Target of Rapamycin Complex 1, TORC1. Our RNA-seq data also highlights that *TCO89*, a TORC1 subunit, is downregulated in a genipin protective scenario (Supplementary table 4), further confirming a genipin interference with central cellular metabolism.

## Endocytosis is enhanced upon genipin treatment

αSyn-GFP overexpression is described to strongly affect endocytosis and vesicular trafficking[19]. Interestingly, these are biological processes for which gene alterations (*YPT1*, *YPK1*, *HRR25*, Supplementary Table 4) were highlighted by our RNA-seq analysis. We then decided to evaluate the effect of genipin in these processes by using FM4-64, a dye that selectively stains yeast vacuolar membranes and enters the cells via the endocytic pathway. αSyn-GFP overexpression promoted a scattered pattern of FM4-64 puncta (Fig. 6a). Importantly, when αSyn-GFP overexpressing cells were incubated with genipin, FM4-64 was efficiently uptaken and delivered to the vacuole, as observed in control cells (Fig. 6a). Moreover, this uptake was endocytosis-mediated, since αSyn-GFP overexpressing cells did not present this phenotype when incubated with the blue-fluorescent chloromethyl derivative of aminocoumarin (CMAC), a dye described to enter cells by diffusion (Supplementary Fig. 5).

*YPT1, HRR25 and YPK1* were genes highlighted by our RNA-seq analysis (Fig. 4d). Ypt1, is a Rab GTPase which controls the first steps of Endoplasmic Reticulum-to-Golgi (ER-to-Golgi) trafficking in yeast cells. Moreover, casein kinase 1 (CK1) family member Hrr25 (Fig. 4d) has been shown to contribute to the delivery of ER-derived vesicles to the Golgi[20] and is regulated by Ypt1. Also, Ypk1 is a serine/threonine kinase required to control endocytosis by phosphorylating components of the endocytic machinery. The impact of genipin protection against αSyn-GFP effects on these genes was further confirmed by RT-qPCR. Importantly, *YPT1* was downregulated (Fig. 6b) and *HRR25* and *YPK1* were upregulated (Fig. 6c, d) in αSyn-GFP overexpressing cells treated with genipin (the protective scenario), which is in line with increased

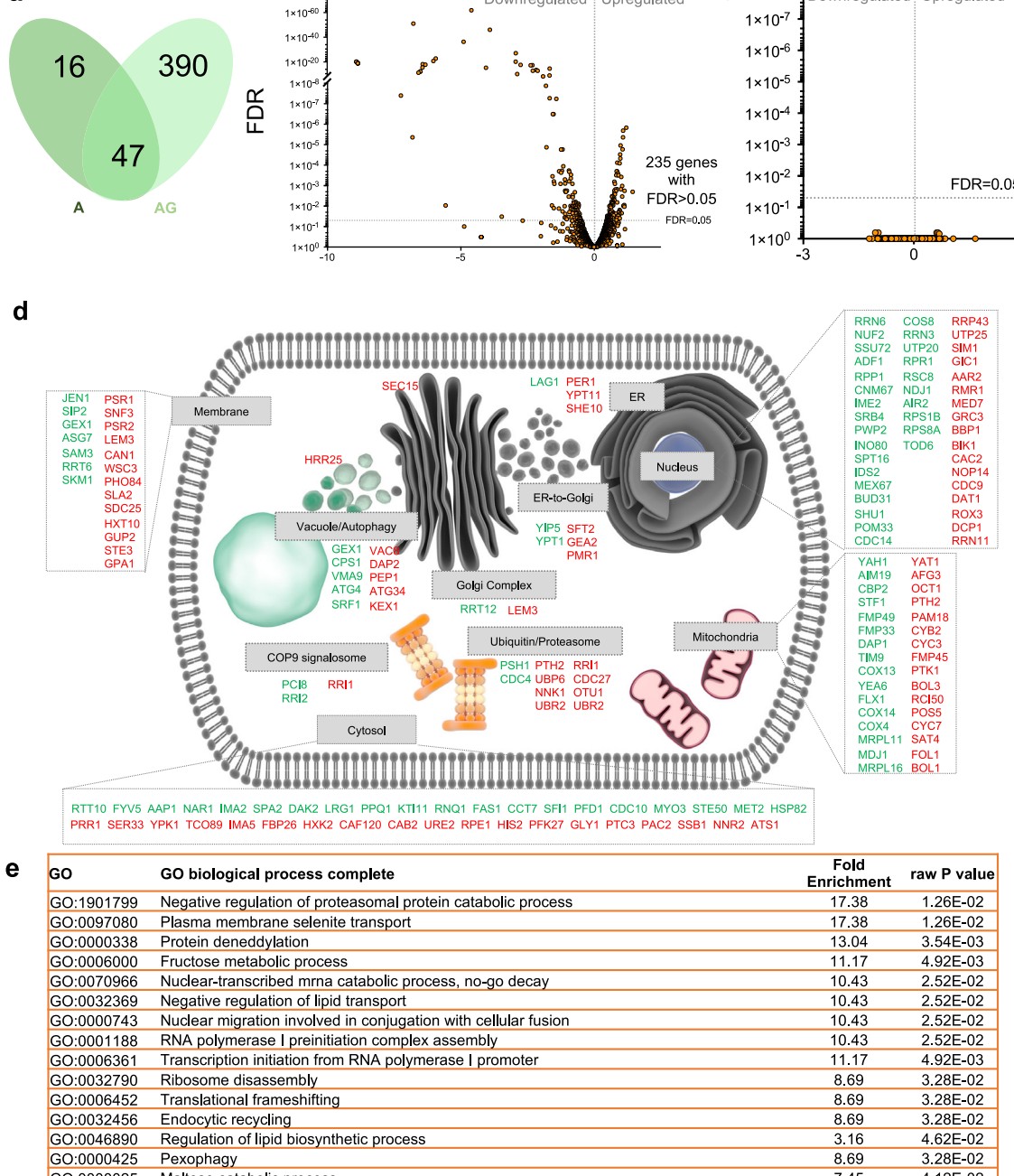

**e**

| GO | GO biological process complete | Fold Enrichment | raw P value |
|---|---|---|---|
| GO:1901799 | Negative regulation of proteasomal protein catabolic process | 17.38 | 1.26E-02 |
| GO:0097080 | Plasma membrane selenite transport | 17.38 | 1.26E-02 |
| GO:0000338 | Protein deneddylation | 13.04 | 3.54E-03 |
| GO:0006000 | Fructose metabolic process | 11.17 | 4.92E-03 |
| GO:0070966 | Nuclear-transcribed mrna catabolic process, no-go decay | 10.43 | 2.52E-02 |
| GO:0032369 | Negative regulation of lipid transport | 10.43 | 2.52E-02 |
| GO:0000743 | Nuclear migration involved in conjugation with cellular fusion | 10.43 | 2.52E-02 |
| GO:0001188 | RNA polymerase I preinitiation complex assembly | 10.43 | 2.52E-02 |
| GO:0006361 | Transcription initiation from RNA polymerase I promoter | 11.17 | 4.92E-03 |
| GO:0032790 | Ribosome disassembly | 8.69 | 3.28E-02 |
| GO:0006452 | Translational frameshifting | 8.69 | 3.28E-02 |
| GO:0032456 | Endocytic recycling | 8.69 | 3.28E-02 |
| GO:0046890 | Regulation of lipid biosynthetic process | 3.16 | 4.62E-02 |
| GO:0000425 | Pexophagy | 8.69 | 3.28E-02 |
| GO:0000025 | Maltose catabolic process | 7.45 | 4.12E-02 |
| GO:0051726 | Cell cycle arrest | 7.45 | 4.12E-02 |
| GO:0032447 | Protein urmylation | 7.45 | 4.12E-02 |
| GO:0006123 | Mitochondrial electron transport, cytochrome c to oxygen | 6.13 | 6.61E-03 |
| GO:0032482 | Rab protein signal transduction | 6.02 | 1.95E-02 |
| GO:0015880 | Coenzyme transport | 5.59 | 2.31E-02 |
| GO:0000750 | Pheromone-dependent signal transduction involved in conjugation with cellular fusion | 5.43 | 3.78E-03 |
| GO:0046835 | Carbohydrate phosphorylation | 5.21 | 2.69E-02 |
| GO:0007121 | Bipolar cellular bud site selection | 4.35 | 4.05E-02 |
| GO:0071472 | Cellular response to salt stress | 4.35 | 4.05E-02 |
| GO:0015755 | Fructose transmembrane transport | 4.35 | 4.05E-02 |
| GO:0000920 | Septum digestion after cytokinesis | 4.35 | 4.05E-02 |
| GO:0051085 | Chaperone cofactor-dependent protein refolding | 4.12 | 4.56E-02 |

**Fig. 4 | Genipin induces differential gene expression and it is dependent on αSyn-GFP expression. a** Venn diagram of genes differentially expressed in αSyn-GFP expressing cells comparing with cells expressing the empty vector and αSyn-GFP expressing cells treated with genipin comparing with cells expressing the empty vector. **b** Volcano plot of differentially expressed genes in αSyn-GFP expressing cells treated with genipin comparing with αSyn-GFP expressing cells. **c** Volcano plot of differentially expressed genes in cells expressing the empty

vector treated with genipin comparing with cells expressing the empty vector. **d** Illustration (built with the free library of science and medical illustration from somersault18:24) of the distribution of the annotated genes differentially expressed in αSyn-GFP expressing cells treated with genipin comparing with αSyn-GFP expressing cells (Supplementary table 4). **e** Gene Ontology (GO) analysis of Biological Processes enriched upon the incubation of αSyn-GFP expressing cells with 10 µM of genipin. αSyn-GFP cells – A, αSyn-GFP cells + 10 µM genipin – AG.

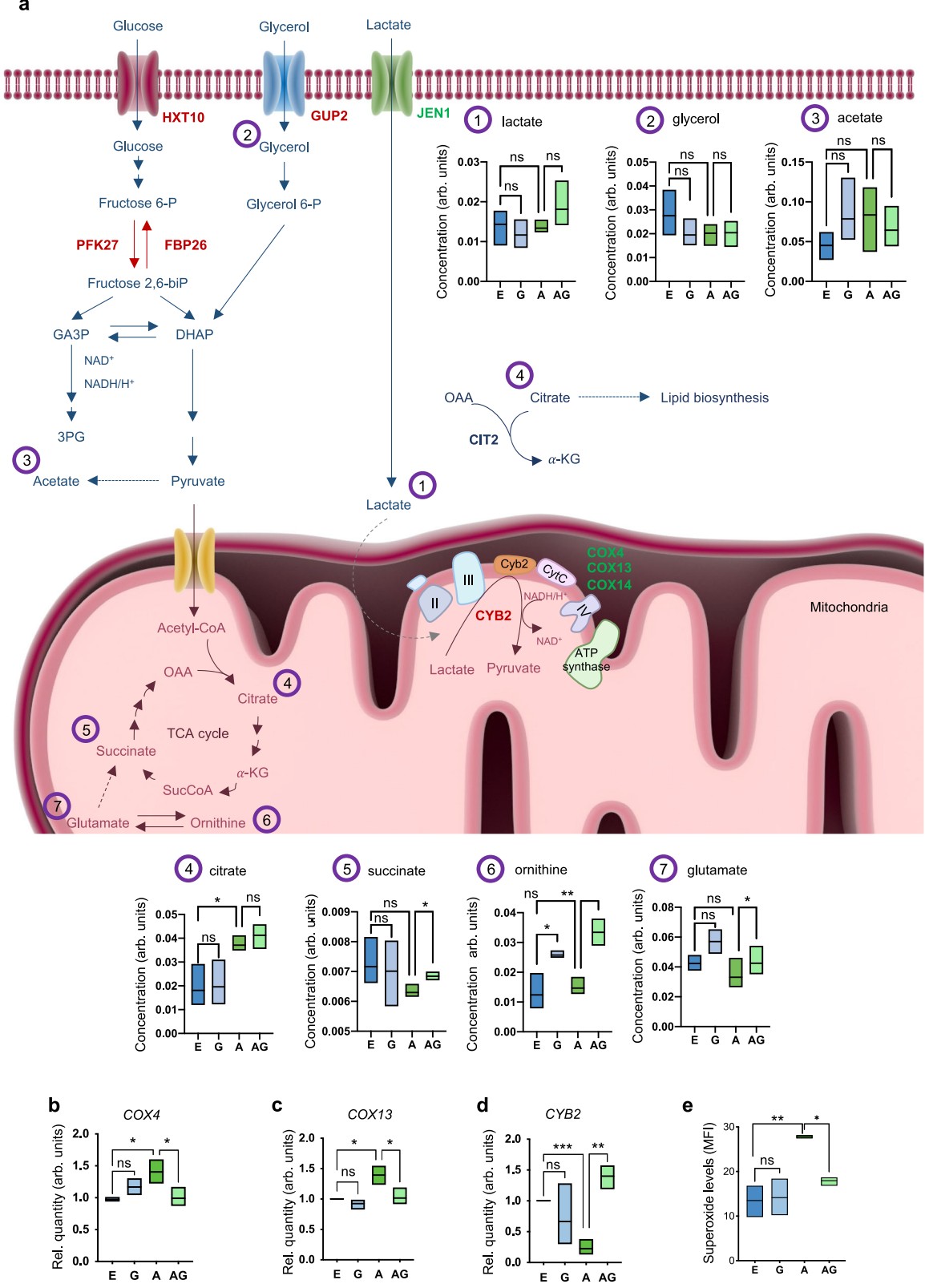

FM4-64 uptake, suggesting that genipin ameliorates endocytosis and vesicular trafficking, protecting yeast cells from αSyn-GFP toxicity.

## Genipin targets lipid homeostasis to protect yeast cells

αSyn plays a role in synaptic vesicle recycling, synthesis, and vesicular storage. These processes are dependent on lipid homeostasis and, in fact, αSyn is proposed to play a role in lipid transport, packing and membrane biogenesis[21]. A tight control of lipid pathways is of extreme importance not only for these processes but also to maintain a stable cellular environment.

We observed that αSyn membrane interaction sites are significantly affected by genipin as observed by the reduction of NMR signal in the N-terminal (Fig. 3f–h) and that lipid biosynthesis and transport are processes targeted by genipin in αSyn-GFP

**Fig. 5 | Genipin promotes metabolic alterations in αSyn-GFP overexpressing cells. a** Overview of genes from metabolic pathways differentially expressed in αSyn-GFP overexpressing cells treated with genipin when comparing with cells overexpressing αSyn-GFP, assessed by RNA-seq. Glucose, lactate and glycerol transporters genes (*HXT10, JEN1, GUP2*, respectively); *PFK27* and *FBP26* genes from glycolysis; Complex IV genes *COX4, COX13* and *COX14*; Alternative cytochrome c gene, *CYB2;* Citrate synthase, *CIT2*. Genes denoted in green and red are down-regulated and upregulated, respectively. Metabolites were quantified by ¹H-NMR and are total intracellular levels (*n* = 3). Compound 1 - 7: 1 – lactate, 2 – glycerol, 3 – acetate, 4 – citrate (E vs A *$p$ = 0.0332), 5 – succinate (A vs AG *$p$ = 0.0313), 6 – ornithine (E vs G *$p$ = 0.0221; A vs AG **$p$ = 0.0032), 7 – glutamate (A vs AG *$p$ = 0.0159). Fructose-6P – Fructose 6-phosphate, Fructose-2,6-biP – Fructose 2,6-biphosphate, GA3P – Glyceraldehyde 3-phosphate, DHAP – Dihydroxyacetone phosphate, 3PG – Glycerate 3-phosphate, Glycerol-3P – Glycerol 3 phosphate, α-KG – α-Ketoglutarate, SucCoA – Succinyl CoA, OAA – Oxaloacetate. **b** COX4 (E vs A *$p$ = 0.0177; A vs AG *$p$ = 0.0433, $n$ = 3), **c** COX13 (E vs A *$p$ = 0.0123; A vs AG *$p$ = 0.0398, $n$ = 3) and **d** CYB2 (E vs A ***$p$ = 0.0006; A vs AG **$p$ = 0.0010, $n$ = 3) mRNA levels were assessed by RT-qPCR. **e** Superoxide levels were assessed by FCM using DHE staining (E vs A $p$ = 0.0035; A vs AG $p$ = 0.0282, $n$ = 3). Control cells with empty vector – E, Control cells + 10 μM genipin – G, αSyn-GFP cells – A, αSyn-GFP cells + 10 μM genipin – AG.

overexpressing cells (Fig. 4e). Therefore, we further investigated if genipin modulates αSyn-GFP's interaction with the cell membrane by assessing its capacity to use the cell membrane as a nucleation site for further aggregation.

αSyn-GFP preferentially localizes and nucleates in yeast plasma membranes, leading to inclusion formation. That event is described to interfere with several processes leading to toxicity[4,5]. Upon genipin incubation, the percentage of cells with plasma membranal inclusions decreases, while the percentage of cells with cytoplasmic inclusions remains unaltered (Fig. 6e, f). This result further supports the hypothesis that genipin decreases the nucleation of αSyn-GFP in the plasma membrane, preventing its aggregation.

This led us to further investigate how lipid homeostasis and lipid storage alterations could interfere with genipin protection. We measured the levels of neutral lipids by FCM and found that, upon αSyn-GFP overexpression, the total amount of neutral lipids increases (Fig. 6g), suggesting an increase in lipid storage, a result that has been previously described[19,22]. This increase in neutral lipids and the fact that αSyn has been shown to bind to lipid droplets (LD), contributing to LD homeostasis by regulating triglycerides synthesis and lipolysis, the exact underlying molecular mechanisms are still unknown.

Interestingly, genipin further increases the accumulation of neutral lipids in cells overexpressing αSyn-GFP (Fig. 6g), suggesting a role for lipid metabolism and storage in genipin protection. This effect seems to be completely dependent on the presence of αSyn-GFP due to the lack of effect of genipin in the amount of neutral lipids in control cells.

The RNA-seq data shows that Low Dye Binding protein 16 (*LDB16*) was upregulated in αSyn-GFP overexpressing cells, with or without genipin treatment. This observation was confirmed by RT-qPCR (Fig. 6h). Ldb16, together with seipin (Sei1), is responsible for controlling the formation of lipid droplets and it was described as a key player in αSyn pathophysiology[22]. To evaluate a possible role of Ldb16 in αSyn-GFP overexpressing cells and in genipin treated cells, we evaluated αSyn-GFP aggregation and toxicity in *ldb16* mutant cells.

Interestingly, deletion of *LDB16* abolished the capacity of genipin to decrease both toxicity (Fig. 6i) and the percentage of cells with αSyn-GFP inclusions (Figs. 6j, k), suggesting that lipid droplet formation is needed for genipin protection to occur. The over-expression of αSyn-GFP on *ldb16* mutant resulted in a toxicity phenotype less pronounced than that observed for the respective wild type (WT) strain, the BY4741 (Fig. 6j vs 6 l, 6i vs 6 m). Importantly genipin protection is retained significantly in this background either in rescuing αSyn-GFP toxicity and in rescuing cellular growth (Fig. 6l and Supplementary Fig. 6a) and reducing αSyn-GFP inclusions (Fig. 6m). However, it is interesting that in the BY4741 background strain an exacerbation in αSyn-GFP inclusions seems to occur when *LDB16* is deleted (Fig. 6j). The impact of *LDB16* deletion on FM4-64 distribution after αSyn-GFP overexpression is heterogenous. Although the impairment of FM4-64 uptake was still observed in αSyn-GFP overexpressing cells (like observed for background strain, Supplementary Fig. 6b), a slight vacuole staining

could be detected (Fig. 6o). In addition, genipin improvement of FM4-64 trafficking towards the vacuole was clearly observed (Fig. 6o). To further confirm the impact of genipin on *ldb16* mutant vesicular trafficking, the processing of Carboxipeptidase Y (CPY) protein was assessed. CPY is normally synthetized in the ER (unprocessed CPY, 69 kDa), it is further processed within the Golgi (processed CPY, 67 kDa) and subsequently transported to the vacuole. Upon vesicular trafficking impairment, the ER-form (unprocessed) is retained and this retention was used as surrogate for trafficking assessment. Interestingly, the levels of CPY retained in the ER (unprocessed CPY) increased after αSyn-GFP over-expression, both in WT (Supplementary Fig. 6c, 6d) and in *ldb16* mutant cells (Fig. 6p). Importantly, genipin failed to decrease CPY levels in all conditions (Supplementary Fig. 6c, d and Fig. 6p, q), suggesting that its protective capacity does not depend on vesicular trafficking but rather on endocytosis improvement. Having into consideration the key role of lipids in endocytosis, we further investigated the interplay between lipid homeostasis, membrane composition and genipin protection, using *ERG6* deletion mutant. Erg6 converts zymosterol to fecosterol in the ergosterol biosynthetic pathway and *erg6* mutant accumulates zymosterol but cannot synthesize ergosterol. A reduction in ergosterol production is expected to affect membrane fluidity and integrity and, therefore might modulate membrane-binding affinity of αSyn[23]. As expected, genipin had no impact on the toxicity of *erg6* mutant cells expressing the empty vector (Fig. 6r). Upon the overexpression of αSyn-GFP on *erg6* mutant, there is a trend for an increased toxicity, however not significant. Importantly, genipin was still able to decrease the percentage of cells displaying αSyn-GFP inclusions in this mutant (Fig. 6s, t) comparable to the protection observed in the respective background strain (Fig. 6l–n). Overall, these results suggest that membrane composition in ergosterol do not affect genipin's protective activity.

## Genipin increases lifespan and motor performance in *Drosophila* concomitant with reduction of αSyn aggregation

*Drosophila* models based on αSyn-GFP overexpression mimic several hallmarks of PD, including aggregation, decreased survival and motor deficits[24,25].

To test genipin protection in vivo, we used transgenic over-expression of human αSyn-GFP in *Drosophila*, under the control of ELAV-GAL4, which allowed us to overexpress it in the entire nervous system. Genipin was tested for toxicity in WT flies (Control flies, ELAV-GAL4) and it did not present toxicity at any concentrations tested between 10 μM – 10 mM (Supplementary Fig. 7a). As expected, flies overexpressing αSyn-GFP (PD flies) have decreased longevity, however, when reared in fly food containing 1 mM of genipin, a significant increase in the survival rate was observed (Fig. 7a). Motor disabilities are one of the most striking symptoms of PD, and our *Drosophila* PD model mimics this feature. The overexpression of αSyn-GFP in the flies' nervous system caused motor impairments (Fig. 7b) as evaluated by the climbing index. PD flies suffer from severe motor impairments which can be observed as early as in day

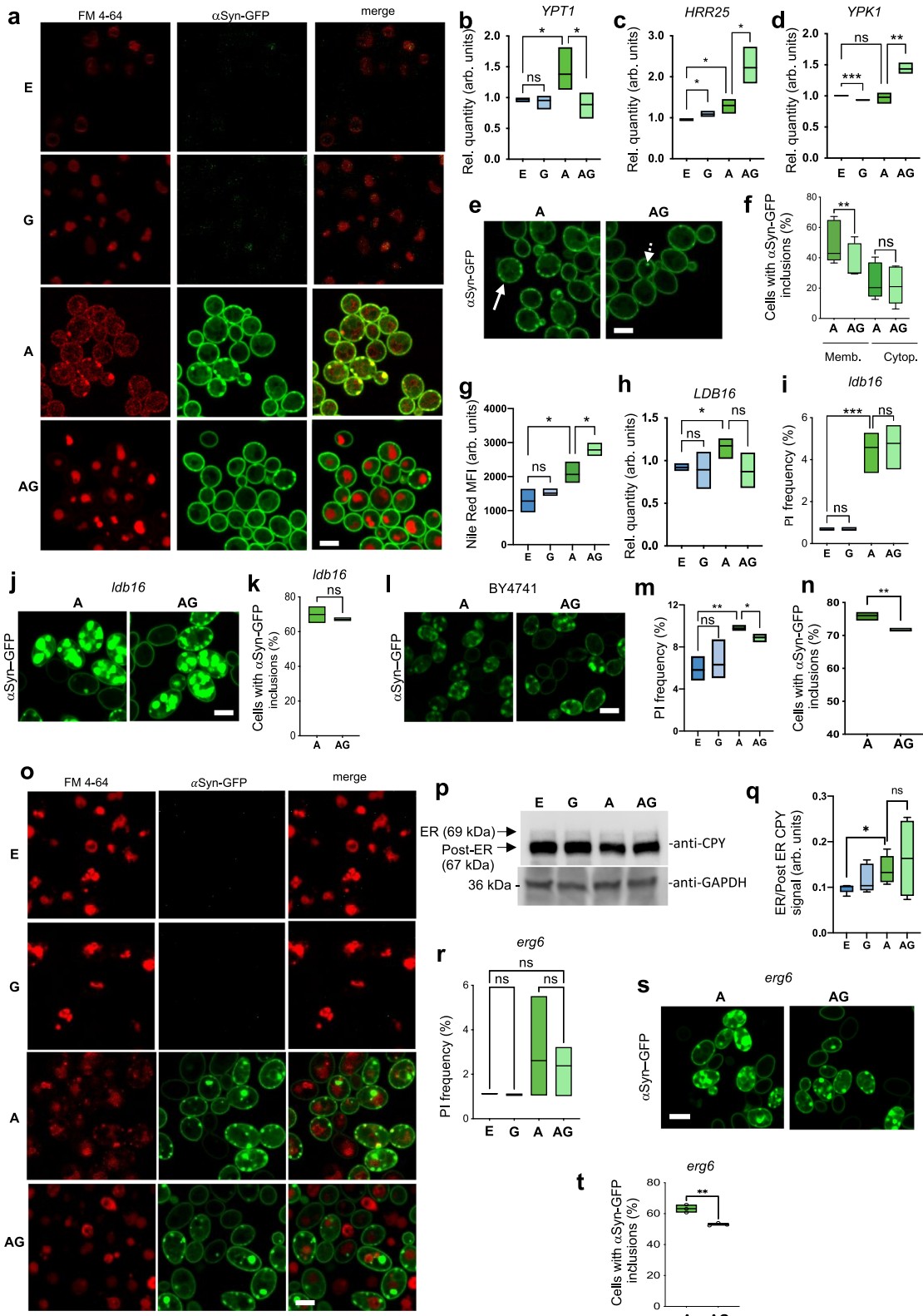

17 (Fig. 7b), as a marked increase in the climbing index that worsens until day 30. Importantly, genipin-treated flies display drastic improvement in motor capabilities (Fig. 7b) demonstrating in vivo that genipin is a protective molecule against αSyn deleterious effects in *Drosophila* brain.

In order to better characterize the mechanism of action of genipin in flies, we analysed its ability to modulate the levels of expression and aggregation of αSyn-GFP in the nervous system. We demonstrate, by

western blotting protein extracts of 17 days old adult heads, that genipin does not affect the levels of expression of αSyn-GFP in the brain (Fig. 7c). Meanwhile, by performing a Triton-X solubility experiment, we found that genipin affects the solubility status of αSyn-GFP in *Drosophila* brains. The analysis of protein extracts from 17 days old adult heads, demonstrates that genipin promotes a reduction in the levels of αSyn-GFP in the Triton-X insoluble fraction (Fig. 7d). This result indicates that the therapeutic effect of genipin observed in flies

**Fig. 6 | Genipin targets endocytosis and lipid storage to protect yeast from αSyn-GFP toxicity. a** Endocytosis evaluation by FM4-64 incubation by confocal microscopy. mRNA levels of **b** *YPT1* (E vs A *$p$ = 0.0446; A vs AG *$p$ = 0.0354, $n$ = 4) **c** *HRR25* (E vs G *$p$ = 0.0458; E vs A *$p$ = 0.0307; A vs AG *$p$ = 0.0327, $n$ = 3) and **d** *YPK1* (E vs G ***$p$ = 0.0004; A vs AG $p$ = 0.0039, $n$ = 3). **e** Yeast cells containing αSyn-GFP inclusions either in the membrane (bulked white arrow) or in the cytoplasm (dashed white arrow) were counted and **f** Plotted (A, 1659 cells; AG, 2044 cells), Membranal inclusions A vs AG, **$p$ = 0.0026 ($n$ = 5). **g** Neutral lipids stained with nile red assessed by FCM (E vs A *$p$ = 0.0279; A vs AG *$p$ = 0.0426, $n$ = 3). **h** *LDB16* mRNA levels (E vs A *$p$ = 0.0350, $n$ = 3). **i** αSyn-GFP toxicity in *ldb16* mutant was assessed by propidium iodide (PI) staining by FCM (E vs A ***$p$ < 0.0001, $n$ = 3). **j** Fluorescence microscopy of *ldb16* mutant αSyn-GFP-overexpressing cells ($n$ = 3). **k** Percentage of *ldb16* mutant cells with αSyn-GFP inclusions. (A, 1249 cells and AG, 1679 cells). **l** Fluorescence microscopy of BY4741 cells overexpressing αSyn-GFP. **m** αSyn-GFP toxicity in BY4741 cells was assessed by PI staining by FCM (E vs A **$p$ = 0.005; A vs AG *$p$ = 0.046, $n$ = 3). **n** Percentage of BY4741 cells with αSyn-GFP inclusions (A vs AG **$p$ = 0.0067, $n$ = 3). **o** *ldb16* mutant endocytosis evaluation by FM4-64 incubation by confocal microscopy. **p** ER accumulation of CPY assessed by western blot and **q** ER/Post ER ratio CPY quantification (E vs A *$p$ = 0.0190, $n$ = 4). **r** αSyn-GFP toxicity in *erg6* mutant was assessed by PI staining by FCM ($n$ = 3). **s** Fluorescence microscopy of *erg6* mutant αSyn-GFP-overexpressing cells ($n$ = 3). **t** Percentage of cells with αSyn-GFP inclusions (A, 2276 cells and AG, 1547 cells; A vs AG **$p$ = 0.0028). Scale bar 5 μm. Control cells with empty vector – E, Control cells + 10 μM genipin – G, αSyn-GFP cells – A, αSyn-GFP cells + 10 μM genipin – AG.

might involve a reduction in the formation of insoluble aggregates containing αSyn-GFP.

Finally, we wanted to investigate whether the expression of αSyn (without a GFP tag) in *Drosophila* adult neurons affected particular kinematic features during walking that could explain the climbing results and if genipin could revert these effects. For this, we used the FlyWalker system, which allows the extraction of kinematic parameters from untethered adult flies with a high spatiotemporal resolution[26,27]. Sustained expression of αSyn increased tarsal contact variability during stance phases and during stance onset relative to the body axis, quantitatively defined by the stance straightness (Fig. 7e, f) and footprint clustering (Fig. 7e, g) parameters, respectively. Moreover, the same animals also displayed a decrease in the use of the tripod gait, the favored leg conformation used by *Drosophila*, with three legs contacting the ground and three in a swing phase[26,28] (Fig. 7h, i), with a corresponding increase in the use of the tetrapod gait, using four legs on the ground (Fig. 7h, j). The same shift was observed in animals exposed to excessive weight bearing[27], or during ageing[29].

The aforementioned kinematic effects of αSyn were partially rescued after genipin treatment. In the presence of genipin, stance traces display an appearance similar to control animals (Fig. 7e, purple brackets), with *stance straightness* values also becoming comparable to control animals (Fig. 7f). *Footprint clustering* values were partially rescued after genipin treatment since these flies are no longer different from the control flies (Fig. 7g). Although genipin-treated animals are also not statistically different from animals expressing αSyn alone in *footprint clustering*, the rescue effect is *on par* with the *stance straightness* results, thus indicating that genipin reduces αSyn-induced motor dysfunction.

Also, flies fed with 1 mM genipin displayed a more pronounced rescue of tripod and tetrapod index becoming, in the case of tripod gait, indistinguishable from control animals (Fig. 7h–j). These results indicate that long-term neuronal expression of αSyn reduces motor efficacy by decreasing leg coordination and tripod usage, an effect partially reversed by feeding 1 mM genipin, consistent with the data obtained in the climbing assay.

## Discussion

At the moment, we still lack disease-modifying therapies for PD and other synucleinopathies. In a time when clinical trials keep failing, the search for new sources of small molecules capable of modulating disease hallmarks is imperative. The discovery of several mutations in *SNCA* gene in PD patients as well as the identification of insoluble aggregated αSyn protein forms as the major component of LBs, the main pathological hallmark of PD, opened a new era in PD research[25,30]. Under normal circumstances, αSyn is soluble and it is enriched at presynaptic terminals, where it is described to bind lipids and to regulate synaptic vesicle release[3,31]. However, αSyn has an increased propensity to aggregate and form oligomeric species. Evidence suggests that αSyn intermediate oligomeric species, rather than the higher-order fibrillar aggregates, are pathogenically toxic and the species responsible for neuronal degeneration and death[32–35]. Also, during this process, there are several cellular events that take place such as mitochondrial injury, oxidative stress, and inflammation, turning PD into a multifactorial disease.

In this study, we aimed to identify the bioactive molecule from a complex mixture previously reported as protective against αSyn pathology and to characterize its mechanism of action.

Importantly, we found that genipin, an iridoid which numerous derivatives were present in the original *C. album* L. PEF leaf extract, provides protection against the deleterious effects of pathological αSyn in yeast and *Drosophila* PD models. Indeed, genipin was previously described as a potential bioactive compound with several effects[36], including neuroprotective[37–41], anti-inflammatory[42–45] and anti-depressant actions[46]. Noteworthy, genipin was never tested for its activity against αSyn aggregation. By taking advantage of the well characterized yeast model of αSyn aggregation, we were able to show that genipin modulates αSyn aggregation in vivo, not only by a direct interference with the protein as demonstrated in vitro, but also by modulating cellular pathways affected by the aggregation process. Bioactivity against αSyn pathophysiology is, indeed, reported for several small molecules. In a screening of 115,000 compounds library, four of them were found to reduce αSyn mediated toxicity in yeast by improving ER-to-Golgi trafficking and decreasing mitochondrial damage[47]. Two other libraries, comprising about 10,000 compounds, were also screened using αSyn overexpression in yeast[48]. (Poly)phenols as quercetin and (−)-epigallocatechin-3-gallate (EGCG) were found to increase cell growth in yeast overexpressing αSyn[48].

In vitro, we showed that genipin strongly impacted on αSyn aggregation process. It was able to decrease the formation of amyloid fibrils and reduce the formation of intermediary oligomeric species. Potent anti-aggregation molecules have been described using in vitro model of αSyn oligomerization, such as 2-nitro-4-(trifluoromethyl) phenyl vinyl sulfone, however, high concentrations were used to see a protective effect (100 μM)[49]. Moreover, eleven synthetic compounds inspired by the structure of known anti-aggregation molecules were also tested for inhibition of αSyn aggregation. Once again, few compounds promoted the decrease in ThT signal and the effect was modest even at high concentrations[50]. Noteworthy, these studies revealed the capacity of non-naturally occurring compounds to modulate αSyn aggregation at high concentrations (10 times higher than genipin used in our study), further reinforcing the potency of genipin.

From a structural point of view, genipin was found to interfere with the first 100 residues of the *N*-terminal, mostly with the first 60, which impacts αSyn oligomerization in a time-dependent manner. The NAC domain (residues 61-95) is then probably found in a more dynamic part of the oligomer while the C-terminal domain remains flexible. Also, genipin was found to interact with regions rich in lysines which can suggest a genipin-mediated cross-linking of αSyn lysines. Actually, it was already described that the ability of small molecules to interact with mutant A53T αSyn lysines (lysine 23 and 33) resulting in a decreased αSyn species' toxicity[51].

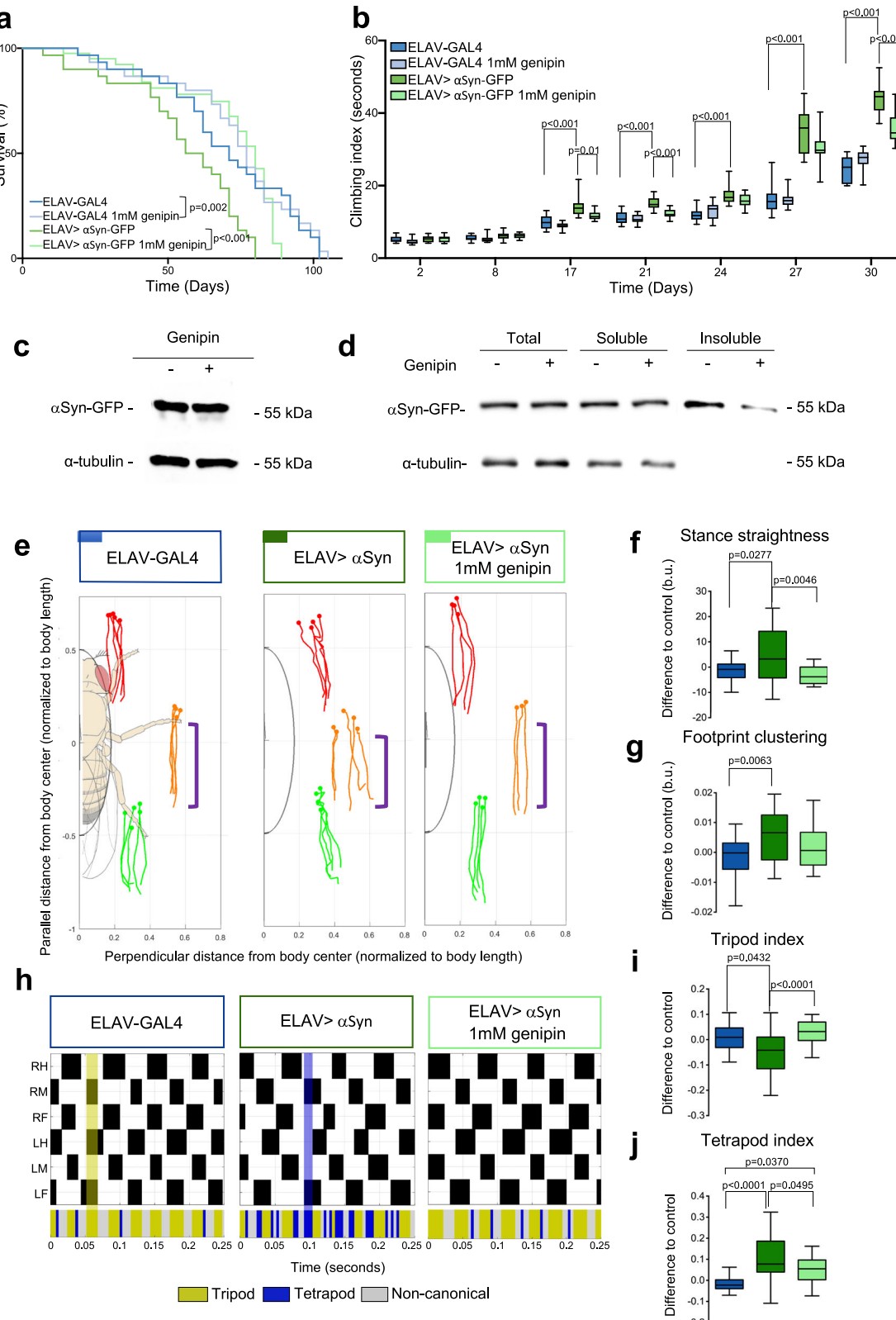

Direct interaction with αSyn was already described for another synthetic small molecule, SynuClean-D (2-hydroxy-5-nitro-6-(3-nitrophenyl)-4-(trifluoromethyl)nicotinonitrile), considered a strong inhibitor of αSyn aggregation[52]. Moreover, SynuClean-D was found to interfere with αSyn fibrils rather than monomers in a region between residue 53 and 73. EGCG effect on αSyn is well characterized and, as here described for genipin, it blocks in vitro fibril formation[53,54]. It was also found that EGCG pushes αSyn to form off-pathway aggregates that are less toxic[55]. Moreover, EGCG was described to inhibit αSyn aggregation by binding to monomer and inhibiting β-sheet formation[55]. NMR analysis suggests that the central hydrophobic region in αSyn is not an interaction site for EGCG since the resonance remains visible. In fact, the affected resonances were spread through αSyn sequence without any apparent correlation with amino acid type,

**Fig. 7 | Genipin protects against the severe motor dysfunction and improves the life span of flies expressing αSyn-GFP in the nervous system. a** Lifespan assessment of flies expressing human WT αSyn-GFP fed or not with 1 mM genipin. Statistical differences accessed by Mantel-Cox test. **b** Climbing assay on flies expressing human WT αSyn-GFP fed or not with 1 mM genipin. All four groups were compared directly and statistics were performed by two-way ANOVA using a matching factor where each row represents a time point. Multiple comparisons were performed using single effects on each row. **c** αSyn-GFP protein levels in the brain of animals **d** and triton-X soluble and insoluble protein fractions assessed by western blot. **e–j** Kinematic data from 24 day-old animals expressing WT αSyn and fed with 1 mM genipin. Blue, ELAV-GAL4 control; Dark green, ELAV-GAL4,UAS-αSyn; Light green, ELAV-GAL4,UAS-αSyn + 1 mM genipin. **e** Representative stance traces. Purple brackets show stance traces corresponding to the tarsal contacts during stance phases relative to the body axis. **f** Stance straightness **g** Footprint clustering. **h** Tripod/tetrapod phases of representative animals walking at similar speeds. Upper section, for each leg swing phases are represented in black (from top to bottom: right hind (RH); right middle (RM); right front (RF); left hind (LH); left middle (LM); left front (LF)). Shaded regions in green and blue show a representative leg combination associated with tripod and tetrapod, respectively. Lower section represents the periods associated with tripod (Yellow), tetrapod (Purple), and non-canonical (Grey) configurations. **i** Tripod index **j** Tetrapod index. Boxplots represent the median as the middle line, with the lower and upper edges of the boxes representing the 25 and 75% quartiles, respectively; the whiskers represent the range of the full data set, excluding outliers. Data was residually normalized and expressed as the difference to ELAV-GAL4 control.

suggesting that EGCG does not interacts in a specific area of αSyn but rather binds randomly to the backbone groups of the protein[55]. On the contrary, genipin presents clear and localized interactions with regions rich in lysines along the folding-prone N-terminal and in the aggregation-prone NAC domain as discussed. Importantly, N-terminal is described to be the folding-prone region of αSyn and is, as well, described to be responsible for membrane interaction. It is also described that the disruption of the highly hydrophobic sequence between residue 71 and 82 (71-VTGVTAVAQKTV-82) abolishes the capacity of αSyn to form amyloid fibrils[56]. Altogether, we can hypothesise that genipin interacts with αSyn structure in a way that it could decrease membrane interaction and binding as well as it could delay or prevent fibril formation. However further structural studies should investigate whether genipin might also be able to disassemble preformed β-sheet-rich structures as well as earlier intermediates of fibrillogenesis through the same type of interactions.

Despite the fact that the exact function of αSyn is yet to be fully understood, it is known that αSyn interacts directly with lipids and membranes, and that the residues within the *N*-terminal are the ones responsible for this interaction[56]. The fact that genipin promotes a strong reduction of NMR backbone signals within the N-terminal region of αSyn, led us to conclude that genipin may interfere with αSyn capacity to interact with membranes. Importantly, it is reported that monomeric αSyn localizes to the membranes at first and then, as a consequence of an increased protein accumulation, it starts to oligomerize culminating in its further aggregation[19]. Therefore, the possible disruption of αSyn interaction with membranes mediated by genipin may culminate in a decreased accumulation of monomers in the membrane. As a consequence, a decreased seeding activity and overall aggregation is then observed by the decreased percentage of cells displaying membranal inclusions.

Having in mind the multifactorial nature of PD, we investigated the possible impact of genipin protection in several central cellular processes affected in αSyn pathophysiology. Based on clinical and experimental PD models, it has been argued that energy and lipid metabolism are among the most affected pathways in PD[18]. In our working model, upon αSyn-GFP overexpression, we detected energy metabolism alterations which are then reversed by genipin.

Under normal conditions of mitochondrial respiration, the reducing equivalents formed in the TCA cycle feed the Electron Transport Chain (ETC). When citrate, an important acetyl donor for fatty acid biosynthesis[57], starts to accumulate in mitochondria, it is transported back to the cytosol where it is cleaved by ATP citrate lyase to form acetyl-CoA and oxaloacetate. In this case, cytosolic acetyl-CoA is used as a building block for lipid biosynthesis. Indeed, citrate levels are found to be increased in αSyn-GFP overexpressing cells and in αSyn-GFP overexpressing cells incubated with genipin. Concomitantly, we also observe an increase of neutral lipids content in αSyn-GFP overexpressing cells, that is potentiated in αSyn-GFP overexpressing cells incubated with genipin.

In the ETC, electrons enter at the level of NADH dehydrogenase and follow oxidative phosphorylation until the final acceptor, $O_2$. The leakage of electrons from the ETC leads to the formation of superoxide. It is well described that increased ROS production is an important hallmark of αSyn pathophysiology. As expected, αSyn-GFP overexpression led to an increase in superoxide levels, as well as to an increase in the expression of genes of the complex IV. Also, we found that genipin decreased the expression of the complex IV genes *COX4*, *COX13*, and *COX14*, suggesting that oxidative phosphorylation might be diminished in the protective scenario. Furthermore, another evidence that the flux of electrons fed to oxidative phosphorylation is decreased is the fact that the levels of some TCA intermediates (succinate) and other metabolites that could feed TCA (ornithine and glutamate) are increased in the protective scenario of genipin. This could be a possible effect of the TCA flux inhibition as the intermediates are accumulating. *CIT2* encodes the non-mitochondrial isoform of citrate synthase that combines citrate with oxaloacetate to form α-ketoglutarate. *CIT2* is described to be induced when the TCA cycle is not working properly[51,58]. The fact that *CIT2* is found to be significantly up-regulated in genipin protective condition, even though the difference to αSyn-GFP cells is not significant, also suggests that the TCA cycle activity is decreased, as proposed. Furthermore, genipin promoted the upregulation of glycolysis and *CYB2*, an alternative cytochrome, which uses lactate as energy source. In fact, total levels of lactate shown a trend to increase upon genipin protection, suggesting the increased availability of lactate for Cyb2. Also, *CYB2* is described to be transcriptionally induced by lactate[59]. *TCO89* (coding for a TORC1 subunit) was downregulated in the genipin protective scenario. The deletion of *TCO89* is reported to increase cell resistance to heat and $H_2O_2$ and to promote cell survival[60]. Interestingly, *tco89* mutants are long-lived mutants shown to undergo a metabolic shift towards alternative carbon sources. These long-life mutants revealed a consistent downregulation of mitochondrial genes such as *COX4* and *COX13*, upregulation of the glucose transporter *HTX1*, upregulation of glycolytic/fermentative genes among them *PFK2*[60], an effect that we also observe in the *alias PFK27*. Altogether, these results suggest that genipin interferes with yeast metabolism culminating in the decrease of oxidative stress, which may exert a role in the improvement of cell survival.

The binding of αSyn to lipids has been extensively characterized in a number of in vitro lipid-based systems[61]. Importantly, lipid accumulation and vesicular trafficking defects have been described upon the overexpression of αSyn in yeast cells[19,22]. In our study, we show that genipin prevented αSyn-damaged endocytosis and increased lipid storage suggesting that its protective activity may be related with these mechanisms. We described that cells expressing αSyn-GFP show disrupted endocytosis, as the uptake of FM4-64 is almost abolished. Importantly, genipin treated αSyn-GFP overexpressing cells are found to be able to uptake FM4-64 similarly to control cells, suggesting that endocytosis is restored. Furthermore, transcriptomic analysis revealed that genes involved in endocytosis are affected by genipin's

protection, namely conserved casein kinase, *Hrr25*, and serine/threonine protein kinase, *Ypk1*.

Hrr25 is described to regulate several cellular processes, namely clathrin-mediated endocytosis or ER-to-Golgi vesicle-mediated transport[62]. It has been previously described that the deletion of *HRR25* or the loss of its kinase activity impairs endocytosis, whereas its overexpression increases the number of endocytic sites, suggesting that Hrr25 activity is critical for endocytic site initiation and/or stabilization[62]. Also, Ypk1 is a serine/threonine kinase which is described to be required for endocytosis[63]. On the other hand, Ypt1 is described to be a suppressor of αSyn toxicity by increasing vesicular trafficking[64].

Importantly, *HRR25* is upregulated in response to αSyn-GFP overexpression and further increased when incubated with genipin, suggesting that Hrr25 potentiation may play a role in alleviating the endocytosis impairment. Noteworthy, *YPK1* is upregulated in cells overexpressing αSyn-GFP and treated with genipin, suggesting that Ypk1 may also play a role, together with Hrr25, in alleviating the endocytosis impairment upon genipin treatment. In fact, it is described that when αSyn-GFP is expressed *in ypk1* mutant, an enhanced αSyn-mediated toxicity is observed, compared with WT[65]. This suggests that Ypk1 enhanced expression by genipin should promote a relief of αSyn-mediated toxicity. However, as observed, genipin incubation does not led to an overexpression of *YPT1* nor to an increase of CPY processing, suggesting that genipin protection is not mediated by an improvement of vesicular trafficking. Fanning and collaborators described that the prevention of lipid droplet formation, either by the deletion of related genes or by a pharmacological approach, could increase αSyn toxicity[22]. Also, they proposed a possible compensatory mechanism where αSyn mediated lipid dis-homeostasis is initially compensated by an increase in lipid storage and therefore protecting cells from αSyn toxicity. Regarding genipin protection, we can infer that this could be a possible mechanism since genipin promoted an increase in neutral lipids and, upon the deletion of *LDB16* (encoding a protein that regulates the formation and size of lipid droplets), it loses its protective capacity. Lipids, such as phosphoinositides, sphingolipids, and sterols, are described to be also central players in endocytosis. Furthermore, ergosterols are the most abundant lipids in yeast membranes. Cells lacking specific proteins in the ergosterol biosynthetic pathway are described to display distinct endocytosis impairments in part due to membrane composition alterations, depending on the gene that is deleted in the pathway and the type of sterols formed[66]. For instance, cells lacking Erg6 accumulates a distinct set of sterols that differ from ergosterol which leads to changes in the membrane composition, however, with little to no effect on endocytosis, contrary to what happens in cells lacking *ERG2*[66]. αSyn-GFP expression in *erg6* cells led to the formation of αSyn-GFP inclusions in a percentage of cells similar to what was verified for WT strain. Importantly, genipin incubation still led to a decrease in the percentage of cells displaying inclusions suggesting that alterations in membrane composition promoted by the deletion of ERG6 are not affecting the protective capacity of genipin, and reinforcing that genipin effect could be related with the improvement of endocytosis.

Importantly, we also verified that genipin can exert its protective activity in vivo, using a previously described *Drosophila* PD model[24]. The overexpression of αSyn-GFP in the entire nervous system was associated with a decrease in the lifespan and motor capabilities of the flies[24,67]. Notably, our study shows that in flies expressing αSyn-GFP fed with genipin there was an increase in lifespan and improved motor capabilities. A comparable effect was observed in flies fed with non-natural chemically synthesized 2-pyridone compounds, an anti αSyn aggregation molecule[67]. Moreover, we were able to demonstrate genipin's ability to modulate αSyn-GFP aggregation in *Drosophila* nervous system.

In this study, we prove that bioactivity-guided fractionations approaches are effective strategies to identify bioactive molecules present in complex mixtures. Importantly, we identified the iridoid genipin with a protective activity against αSyn aggregation and toxicity. We demonstrated that genipin protects against the deleterious effects of αSyn aggregation and toxicity by impacting directly on αSyn structure and, also, on cellular processes affected by αSyn pathophysiology as energy metabolism, lipid storage, or vesicular trafficking. Furthermore, we showed the efficacy of the iridoid genipin in an animal model of PD. These findings pave the way for the exploitation of genipin in mammalian preclinical models and as a lead molecule for PD therapeutics. Moreover, while we strongly suggest that genipin provides a potential lead molecule for PD therapy, it would be interesting to explore the chemical diversity of this class of compounds or modify its chemistry to search for improved lead molecules.

## Methods

### Plant material and extraction procedure

Leaves of *C. album* L. were collected by random sampling in an extensive area of Comporta (a southern region of Portugal). *C. album* samples (collected in Comporta area in 2007) and representative vouchers (1066/2007) were authenticated by Dalila Espírito-Santo, Instituto Superior de Agronomia and deposited on Herbário "João de Carvalho e Vasconcelos", Instituto Superior de Agronomia, Lisbon. Plant material was extracted as follows. Briefly, to each 1 g of *C. album* L. lyophilized leaves, 12 mL of hydroethanolic solvent (ethanol 50% (v/v)) was added (materials at Supplementary table 6). After 30 min of shaking, at room temperature in the dark, the mixture was then centrifuged at 12,400 $g$ for 10 min at room temperature. The supernatant was filtered through a paper filter and then through 0.20 μm cellulose acetate membrane filters. The resulting extracts were stored frozen at −80 °C until further analysis[7].

### Total phenolic quantification

Determination of total phenolic compounds was performed by the Folin-Ciocalteu method adapted to a microplate reader. To each well of a microplate, 235 μL water, 5 μL sample (or solvent, in the control), 15 μL Folin-Ciocalteu's reagent (Fluka®) and 45 μL saturated $Na_2CO_3$ were added. The microplate was incubated for 30 min at 40 °C and the absorbance at 765 nm was measured. Gallic acid was used as standard and the results are expressed in mg of gallic acid equivalents per g of dry weight of plant material (mg GAE.g$^{-1}$ dw)[68].

### *Corema album* L. leaves PEF Fractionation

Lyophilized PEF extract was resuspended with 30 mL of water (each falcon), vortexed and homogenized. The solution was divided into 10 times 15 mL-falcons. The resuspended extract was filtered using 0.22 μm filters in order to prevent insoluble debris from being injected into the column.

The stainless steel (316 L) column was inhouse custom made[69], with 2 cm internal diameter and a length of 10 cm with endcaps with a 10 μm frit (Upchurch), giving an internal volume of 31 ml packed with 31 mL of a C18 silica resin (particle size 55–105 μm, pore size 125 Å). 0.1% Formic acid in miliQ water was chosen as solvent A and 100% methanol as solvent B, in a Akta Explorer 10[70] (Cytiva, Uppsala, Sweden) Sugars were present only in the flow through, which was discarded.

For fractionation, 10 mL of the sample was injected followed by a washing step using 31 mL (1 Column Volume – 31 mL) of solvent A, a gradient elution from 0 to 10% solvent B in 10 mL, a gradient elution from 10 to 68.5% solvent B in 364 mL, a gradient elution plus a regeneration step with 100% solvent B for 50 mL. A flow rate of 3 mL.min$^{-1}$ was used in all the steps.

A total of 16 fractions were collected from the fractionation. Equivalent fractions from different fractionations were combined. A

**Table 1 | Description of strains and plasmids used**

| Type | Name | Genotype |
|---|---|---|
| Strain | Control cells[72] | W303 MAT a; can1-100 his3-11 15 leu2-3 112 pRS304::TRP1 pRS306::URA3 ade2-1 |
| Strain | αSyn-GFP cells[72] | W303 MAT a; can1-100 his3-11 15 leu2-3 112 GAL1pr-SNCA(WT)-GFP::TRP1 GAL1pr-SNCA(WT)-GFP::URA3 ade2-1 |
| Strain | 2x FUS[73] | W303 MATα; can1-100, GAL1pr-FUS::HIS,15, leu2-3,112, GAL1pr-FUS::TRP, ura3-1, ade2-1 |
| Strain | BY4741 | MAT a; his3Δ1; leu2Δ0; met15Δ0; ura3Δ0 |
| Strain | ldb16 | MAT a; his3Δ1; leu2Δ0; met15Δ0; ura3Δ0; Δldb16::KanMX4 |
| Strain | erg6 | MAT a; his3Δ1; leu2Δ0; met15Δ0; ura3Δ0; erg6::KanMX4 |
| Plasmid | Htt | p426_GAL-Htt exon 1-GFP |
| Plasmid | Empty vector[19] | p426_GAL |
| Plasmid | αSyn -GFP 2μ[19] | p426_GAL-SNCA(WT)-GFP |
| Plasmid | uGFP[7] | p413_GPD-uGFP |

total of 8 fractionations were performed in this manner. Each fractionation included 9.5 mg GAE total phenols concentration, which counts for a total of 76 mg GAE. All fractions were evaporated using a rapidvap with cold trap (Labconcon, USA) for 1 h, at a pressure lower than 5 mbar to ensure the total methanol evaporation. The fractions were stored at −80 °C until further analysis. The fractionation profile in terms of total phenols and solvent % was represented using Microsoft Excel (v 16.69.1).

## Phenolic profile determination by LC–MS
The 16 *C. album* L. fractions were resuspended in 1 mL of 50:50 $H_2O$: acetonitrile with 0.1% formic acid, subsequently diluted 5-fold with $H_2O$, and filtered using 0.45 μm filter vials (Thomson Instrument). Samples were then analyzed as described by Kallscheuer et al.[71]. In summary, samples were analyzed by LC-MS comprised of an Accela 600 quaternary pump and an Accela PDA detector coupled to a linear trap quadrupole (LTQ) Orbitrap XL mass spectrometer (Thermo Fisher Scientific) with accurate mass capabilities, operated under the Xcalibur software package 2.0. Five microliters of the samples were injected into a Synergi Hydro RP-80å 150 ×2 mm, particle size 4 μm (Phenomenex) and autosampler, and column temperatures were maintained at 4 °C and 30 °C, respectively. Samples were eluted at a flow rate of 0.3 mL min using two mobile phases (A: 0.1% v/v formic acid in $H_2O$; B: 0.1% formic acid in acetonitrile) with the following gradient: 0 min: 2% B; 2 min: 2% B; 5 min: 5% B; 25 min: 45% B; 26 min: 100% B; 29 min: 100% B; 30 min: 2% B; 35 min: 2% B. Mass detection was carried out in both positive and negative ESI modes. Data acquisition was performed in ESI full scan mode within the FT detector operating with a mass resolution of 30.000 (full width at half-maximum defined at m/z 400); the HESI probe temperature was set to 100 °C, the capillary temperature at 275 °C with sheath gas at 60 psi, and auxiliary gas at 30 psi. For the positive polarity, the source voltage was set at +4.5 kV, capillary voltage at 44 V, and tube lens voltage at 100 V. For negative polarity, the source voltage was set at −3.5 kV, capillary voltage at −44 V, and tube lens voltage at −100 V. In addition, the analysis was repeated in full-scan mode within the FT detector and followed by a data-dependent MS/MS of the three most intense ions within the LTQ ion trap detector using Helium as a collision gas, with a normalized collision energy of 45, mass isolation width of ±1 m/z Activation Q of 0.25 and activation time of 30 ms, source voltage (set at 3.4 kV) in wide band activation mode. Full scan FT data are acquired in profile mode, whereas the DDA MS2 fragmentation spectra are acquired in centroid mode. A scan speed of 0.1 s and 0.4 s are applied in the LTQ and the Fourier transform mass spectrometer, respectively. The Automatic Gain Control was set to $1 \times 10^5$ and $5 \times 10^5$ for the LTQ and the Fourier transform mass spectrometer, respectively. The dataset was analysed using Xcalibur and peaks present in the bioactive fractions were tentatively annotated by utilizing the accurate mass (<3 ppm), MS/MS fragmentation patterns, and reference to retention time for previously analyzed

standards when available (Supplemental Table 2). Briefly, accurate mass was used to generate molecular formulas, which were subsequently filtered down according to the likelihood of occurring in nature (i.e. C/H ratios, heteroatom ratio check and element probability check), the $^{13}C$ isotope ratio observed, the type of adducts formed and the polarity of the molecule relative to the chromatographic separation (for example lipids would be unlikely to elute in highly polar sections of the separation). Subsequently the potential molecular formulas were queried in multiple compound databases (Chemspider and METLIN) and our in-house database to identify potential compound matches. These were then filtered down based on the fragmentation pattern observed in our experimental data.

## Yeast plasmids, strains, and transformation
Plasmids and strains used in this study are listed in Table 1.

## Yeast growth conditions
Synthetic complete (SC) media [0.67% (w/v) yeast nitrogen base without amino acids (YNB) (Difco) and 0.79 g $L^{-1}$ complete supplement mixture (CSM) (Qbiogene, Montreal, QC, Canada) or lacking specific amino acids, containing 1% (w/v) raffinose, 2% (w/v) galactose, or 2% (w/v) glucose were used.

For all experiments, a pre-inoculum was prepared in raffinose medium and cultures were incubated overnight at 30 °C under orbital shaking. Cells were diluted in fresh medium and incubated under the same conditions until the optical density at 600 nm ($OD_{600}$) reached 0.6 ± 0.05 (log phase). αSyn-GFP expression was performed by incubation with medium containing 2% galactose with an $OD_{600} = 0.2 ± 0.002$ (early exponential phase).

## PEF fractions and compound testing
For growth curves, yeast cultures were diluted to an $OD_{600}$ of 0.04 ± 0.004 in a medium containing glucose or galactose and supplemented or not with each fraction or with commercially available genipin (Extrasynthese, Genay Cedex, France) in a 96-well microtiter plate. The cultures were then incubated at 30 °C with shaking for 24 h and cellular growth was kinetically monitored hourly by measuring $OD_{600}$. Data was modelled using nonlinear parametric regressions and growth parameters (final biomass, maximum growth rate, lag time, doubling time, and area under the curve) using grofit package as well as % of protection (area under curve relative to the control strain), all with 95% confidence intervals, were estimated from the best-fit model using Rstudio (Rstudio Version 0.99.902, GNU lesser general public license, Boston, MA, USA). Before performing the bioactivity assays, all fractions, and compounds were tested for toxicity in the respective control strains.

To evaluate cell viability by flow cytometry (FCM) with propidium iodide (PI), αSyn-GFP expression was induced during 6 h and cells were incubated or not with 10 μM of genipin.

For αSyn clearance experiments, after 6 h of αSyn-GFP expression induction, the cells were centrifuged at 2500 g washed in PBS, resuspended in 2% (w/v) glucose media (αSyn expression OFF), and incubated at 30 °C, with shaking, for 24 h. The levels of αSyn-GFP were determined by western blotting at 6 h of induction (corresponding to 0 h of clearance) and at 24 h of clearance.

## Western blotting

For αSyn quantification, total yeast protein extraction was performed using TBS buffer and glass bead lysis in the presence of protease and phosphatase inhibitor cocktail (Roche, Mannheim, Germany)[7,72]. Protein quantification was performed using Micro BCA™ Protein Assay Kit (Thermo Scientific, Rockford, USA). After extraction, a western blot was performed following standard procedures. Antibodies used: 1:5000 αSyn (Purified Mouse Anti-α-Synuclein, Clone 42/α-Synuclein (RUO), Catalog Number 610787, BD Transduction Laboratories, San Jose, CA, USA), 1:5000 PGK (Clone 22C5D8, Catalog number 459250, Life Technologies, Paisley, UK), 1:2500 GAPDH (Clone GA1R, Catalog number MA5-15738, Thermo Scientific, Rockford, USA), 1:500 CPY (Clone 10A5B5, Catalog number A-6428, Life Technologies, Paisley, UK).

For *Drosophila* samples, proteins were extracted in lysis buffer supplemented with Complete Protease Inhibitor Cocktail tablets from Roche (Basel, Switzerland). The protein extracts were obtained from 17 days old flies. Total protein was quantified using the DC Protein Assay, from Bio-Rad (CA, USA). For the western blot, we used anti-GFP (Clone 3H9, Catalog number 3h9-100), from Chromotek and anti-α-tubulin (Clone AA4.3, Catalog number RRID:AB_579793, Developmental Studies Hybridoma Bank, IA, USA).

## Triton-X solubility

The Triton-X solubility experiment in *Drosophila*, was performed using protein extracts obtained from 17 days old flies. Briefly, 200 μg of total protein extract was incubated with 1% Triton X-100 on the ice during 30 min. Triton soluble and insoluble proteins fractions were separated by a 60-min centrifugation step at 15,000 g at 4 °C. The supernatant, containing the soluble proteins fraction (Triton-X soluble), was carefully collected and the pellet, containing the insoluble proteins fraction (Triton-X insoluble), was resuspended in 40 μL of 2% sodium dodecyl sulfate Tris–HCl buffer pH 7.4 by pipetting and sonication for 10 s[73]. For the immunoblotting analysis, equal volumes of each fraction were loaded and the presence of αSyn-GFP in the total, soluble and insoluble fractions was detected using anti-GFP (3H9), from Chromotek. Additionally, anti-α-tubulin (AA4.3) from Developmental Studies Hybridoma Bank (IA, USA) was used as a loading control.

## Flow cytometry

Flow cytometry (FCM) was performed in a CyFlow Cube 6 (Sysmex Partec GmbH, Görlitz, Germany), equipped with the 695/40 BP and the 685 LP. To assess cell viability with PI, yeast cells were incubated with 5 μg.mL$^{-1}$ of PI, for 30 min protected from light, under agitation. For superoxide quantification, cells were incubated with 30 μM DHE (Molecular Probe, Life Technologies), for 15 min at 30 °C, with agitation and protected from light. Data analysis was performed using FlowJo software and a minimum of 100,000 events were collected for each experiment. Results of cell viability and superoxide levels were expressed as frequency of positive cells and median fluorescence intensity, respectively. To assess the proteasome impairment, uGFP construct was used. Results were expressed as median fluorescence intensity, normalized to control cells. Neutral lipids content was assessed using Yeast Control™ - Neutral lipids kit (Sysmex) according to manufacturer instructions. Data analysis was performed using FlowJo software and a minimum of 100,000 events were collected for each experiment.

## Fluorescence microscopy

To determine the percentage of yeast cells with αSyn-GFP inclusions, cells were grown as described above and GFP fluorescence was visualized using a Carl Zeiss LSM 710 confocal microscope after 6 h of αSyn-GFP induction in the presence or absence of 10 μM of genipin. The percentage of cells presenting αSyn-GFP inclusions was then determined by counting at least 800 cells for each treatment using ImageJ software.

To assess the localization of the αSyn-GFP inclusions, cells with inclusions only in the cytoplasm and cells with inclusions in the membrane were counted. At least 300 cells for each treatment were counted using ImageJ software.

Yeast endocytic pathway and vacuole were stained and using FM™ 4-64 Dye[74] and CellView Blue CMAC, respectively. Briefly, after 6 h of αSyn-GFP overexpression, a pulse of 10 μM of CMAC blue was given to cells for 30 min under agitation, at 30 °C, followed by a chase time in fresh medium of 15 min. Regarding FM4-64, a pulse of 20 μM was given to cells for 15 min under agitation, at 30 °C, followed by a chase time in fresh medium of 120 min, at 30 °C. Fluorescence was then visualized using a Carl Zeiss LSM 710 confocal microscope.

## Aggregation assay

Pure human αSyn was purchased from rPeptide (Watkinsville, USA). Pure αSyn was reconstituted using 50 mM Tris–HCl buffer (pH 7.4). For the aggregation assay, the protein solution (70 μM) was mixed with a final concentration of 10 μM of genipin in 50 mM Tris–HCl buffer (pH 7.4). Protein samples were stirred at 900 rpm, at 37 °C in a thermomixer (Eppendorf, Hamburg, Germany).

## Dynamic light scattering

The aggregation process was performed as described above and monitored by Dynamic Light Scattering using a DLS Zetasizer Nano ZS (Malvern Panalytical, Malver, UK). The samples obtained at t = 25 h were analyzed by SDS–PAGE using standard western blotting procedures as described above.

## Size exclusion chromatography

Size exclusion chromatography (SEC) was performed with pure αSyn aggregated or not in the presence of 10 μM of genipin and total protein lysates from yeast cells expressing αSyn-GFP for 6 h, incubated or not with 10 μM of genipin, extracted as described above and centrifuged at 16,000 g for 10 min to remove any insoluble particles. Samples (70 μM of pure αSyn in 200 μL or 2 mg of total protein extract in a final volume of 500 μL) were analyzed using a Superose 6 10/300 GL column (GE Healthcare, Uppsala, Sweden) using a AKTA Purifier UCP 10 (GE Healthcare, Uppsala, Sweden). The samples were eluted with 50 mM ammonium acetate (pH 7.4) at a flow rate of 500 mL.min$^{-1}$ and the UV absorbance was monitored at 280 nm. To estimate the molecular weight of the protein samples, High Molecular Weight gel filtration calibration kit was used (GE Healthcare, Uppsala, Sweden). Fractions of 500 μL were collected, precipitated overnight at 4 °C in 10% trichloroacetic acid (Sigma-Aldrich, St. Louis, USA) washed in acetone three times and resuspended in protein sample buffer (0.5 M Tris-HCl, pH 6.8, Glycerol, 10% SDS, 0.1% Bromophenol Blue), and were resolved by SDS-PAGE.

Aggregated pure αSyn samples were analyzed by SEC as above and the UV absorbance was monitored at 280 nm.

## Transmission electron microscopy

Overall, 5 μL of each sample was incubated on glow-discharged carbon coated copper grids for 2 min before washing 10 times with dH₂O. 2% uranyl acetate was used to negatively stain the samples for 2 min before imaging on the Tecnai G2 80-200kv TEM.

**Table 2 | Primers' sequence used for RT-qPCR, at a final concentration of 5 μM**

| | |
|---|---|
| YPK1_F | GTTCCACCGCCACATAAGGA |
| YPK1_R | GAACTTTGCTGCTGACTCGC |
| YPT1_F | GGGACACTGCAGGTCAAGAA |
| YPT1_R | CTTGCAGCCACATCTTCACG |
| COX13_F | GCCACTGAAAAGCATGCGAA |
| COX13_R | AATCCCTTGGCCATTCGGAG |
| HRR25_F | TGACATTTACCACGGCACGA |
| HRR25_R | ATGAACGGGATTCCCACACC |
| COX4_F | GGTCCTGGTGCTAAAGAGGG |
| COX4_R | ATGGTACCCTTCCTGGACGA |
| LDB16_F | ACCCATGCCTAAGCCCAATC |
| LDB16_R | GCAAGGGGTGATGTAGCAGT |
| CYB2_F | ACAACGAGCCGAAACTGGAT |
| CYB2_R | CCTGCCCACCTGGATGATTT |
| ALG9_F | CGGGAAGCTTGCTCCTGTAT |
| ALG9_F | CTAGCACAGGCAGTGGGAAA |
| PDA1_F | CTGTTGGTCAGGAGGCCATT |
| PDA1_F | GCATGGAACCACCCTTACCA |

## Protein NMR spectroscopy

NMR experiments were measured on a Bruker 700 MHz spectrometer equipped with a 5 mm triple-resonance, pulsed-field z-gradient cryoprobe. Two-dimensional $^1$H,$^{15}$N heteronuclear single quantum coherence (HSQC) experiments were acquired at 15 °C every 4 h in HEPES buffer (50 mM HEPES, 100 mM NaCl, pH 7.4, 0.01% NaN$_3$) with 5% (v/v) D$_2$O and 3 mM genipin. The protein concentration was 30 μM. Spectra were processed with TopSpin 3.6.1 (Bruker) and analyzed using Sparky 3.115 (T. D. Goddard and D. G. Kneller, SPARKY 3, University of California, San Francisco). The combined HN/N chemical shift perturbation was calculated according to $(((\delta HN)^2 + (\delta N/10)^2)/2)^{1/2}$.

## Metabolite quantification by NMR

For the analysis of the endo-metabolome, cell metabolism quenching was adapted from Lourenço et al., 2012[75]. Briefly, one volume of cell culture was immediately added to three volumes of methanol cold solution (60% v/v; T < −50 °C). Cells were pelleted by centrifugation at 2500 $g$ for 6 min at −10 °C. Pellets were washed three times. Eight mL of the methanol cold solution (T < −50 °C) were added to the pellets, vortexed, and pelleted by centrifugation at 2500 $g$ for 3 min at −10 °C.

Metabolites were extracted based on the method previously described. Two mL of glass beads and 4 mL of ethanol (75% v/v) were added to the cell pellets, the cells were then lysed in a bead beater for 30 s, heated for 3 min to 80 °C, two times. The supernatant was decanted and the extraction step repeated a second time with 2 mL of ethanol (75% v/v). The supernatants were mixed, cleared by centrifugation at 16,000 $g$ for 10 min at −10 °C and dried under vacuum.

Dried samples were dissolved in: 600 μL phosphate buffer in D$_2$O (80 mM, pH 7.0) with 2 mM of sodium azide and 0.16 mM of 3-(trimethylsilyl)propionic-2,2,3,3-d4 (TSP), the suspensions were centrifuged at 21,000 $g$ for 5 min at 4 °C and transferred to 5 mm NMR tubes.

NMR spectra were acquired on a Bruker Avance II + 800 MHz spectrometer equipped with a 5 mm TCI H&F/C/N/-D cryoprobe. All 1D 1H were acquired at 298.15 K and using a noesygppr1d pulse program (128 scans, relaxation delay of 4 s, mixing time of 10 ms, spectral width of 20.0237 ppm, 128k points of free induction decay (FID). Processing of spectra was performed with Bruker TopSpin 3.6.2. All FID were multiplied by an exponential function, followed by Fourier Transformation. Spectra were manually phased and baseline corrected. Chemical shifts were adjusted according to the TSP chemical shift at 0.00 ppm. To spectral assignment, 2D NMR spectra were acquired for some samples: $^1$H-$^1$H TOCSY, $^1$H-$^{13}$C HSQC and $^1$H $J$-resolved.

Metabolite identification and quantification were performed recurring to ChenomxNMRsuite8.12. In some cases, two-dimensional NMR spectra were used to confirm metabolite identification.

## Transcriptomic analysis

Total RNA was extracted using E.Z.N.A.® Yeast RNA Kit according to manufacturer instructions using yeast cells expressing αSyn-GFP or the control vector for 6 h, incubated or not with 10 μM of genipin. The RNA was kept at −80 °C until further analysis.

RNA integrity was verified by gel electrophoresis to verify the presence of two distinct bands corresponding to 18 s and 26 s rRNA subunits and minimal to no smearing was detected. The samples were run on an AATI Fragment Analyzer (Agilent, Santa Clara, USA) using a High Sensitivity Total RNA Analysis Chip (Agilent, Santa Clara, USA) to assess RNA purity and concentration.

RNA quality control, library preparation, and sequencing were conducted by the Genomic Unit at Instituto Gulbenkian da Ciência, Oeiras, Portugal. cDNA and the libraries were obtained using SmartSeq2 protocol as previously described[76]. cDNA and library quality was verified using the DNA 1000 Assay Chip on the AATI Fragment Analyzer (Agilent, Santa Clara, USA). Sequencing was performed using NextSeq 1x75bp single-end sequencing kit (Illumina, San Diego, USA). A minimum of 20 millions reads per sample was recorded.

*Saccharomyces cerevisiae* (S288C) was used as the reference genome (BioProject accession no. PRJNA128). Adapter contamination and low-quality and ambiguous nucleotides were trimmed from the remaining reads. The overall quality assessment of each corrected data set was carried out using FastQC[77]. Single-end reads from the Next-Seq500 were mapped against the de novo assemblies with Bowtie2 and SAMtools v.0.1.19 was used for format file conversion[77]. Differential expression analyses were performed with the R package edgeR[78], after identified features with zero reads in every conditions were removed. All transcripts with a false discovery rate (FDR) correction of the p-value lower than 0.05 were considered as significant. The GO analysis were performed using Panther tool. Cells from three independent experiments were used for RNA extraction and subsequent procedures.

## Quantitative real-time PCR

cDNA Synthesis was performed using Roche Transcriptor First Strand cDNA Synthesis kit according to the manufacturer's instructions. cDNA was diluted 1:50, and quantification of mRNA levels was performed in the Light Cycler 480 Multiwell Plate 96 (Roche) using the Light-Cycler 480 SYBR Green I Master Kit (Roche) and the oligonucleotide primers listed in Table 2 at a final concentration of 5 μM. Reactions were performed in duplicate in a final volume of 20 μL. Cycle's threshold (Ct's) and melting curves were determined using Light Cycler 480 software, version 1.5 (Roche), and results were processed using relative quantification method for relative gene expression analysis[79,80]. Gene expression data were normalized using ALG9 and PDA1 as internal controls.

## Drosophila stocks

UAS-αSyn-EGFP (αSyn-GFP) was generated as previously described[24]. ELAV-GAL4$^{C155}$ driver (#458) was obtained from the Bloomington Stock Center (Indiana University, Bloomington, IN, USA), which allows the expression of the protein of interest in the entire nervous system, under the control of the elav promoter. *Drosophila* stocks were maintained at 25 °C on standard cornmeal media in an incubator with a 12 h light/dark cycle.

## Climbing assays and survival assays

Flies were maintained in standard conditions until the assay. For survival assays, thirty adult flies (15 males and 15 females) of the same age were placed in three vials (10 flies per vial –5 males, 5 females) containing fresh food or containing fresh food embedded with 1 mM genipin. Each 3 days the flies were transferred into new vials with the same feeding conditions and the number of living flies was registered. In survival curves values on the Y-axis represent the percentage of flies alive at each time point analyzed.

Motor function was analyzed by climbing assays. Briefly, groups of 10 males of the same age that were fed or not with 1 mM of genipin were placed into 18-cm-long vials, at room temperature for environmental acclimatization. After a 30 min habituation period to the test vials, they were gently tapped to the bottom of the vial. The climbing time when five flies crossed the 15-cm finish line was recorded[24,81]. Three independent groups of males were tested and performed five trials for each time point. All four groups were compared directly and statistics were performed by two-way ANOVA using a matching factor where each row represents a time point. Multiple comparisons were performed using single effects on each row.

## Kinematic analysis

Kinematic experiments were carried out as described below. After eclosion, flies with the appropriate genotype were collected and maintained in fly food containing 1 mM of genipin, replaced every 2 days. Individual flies were placed into a walking chamber and filmed with a Photron (Tokyo, Japan) Mini UX-100 camera using a Nikon (Tokyo, Japan) AF 24-85 mm lens at 250 frames per second (fps). For each condition at least 20 videos were generated from 10 animals. Videos were tracked and kinetic parameters extracted, using the Fly-Walker software package[26]. Kinematic data was extracted using the FlyWalker Software where each video generates a single data point. Stance linearity is measured by the average difference between the stance traces generated by each leg during stance phase and a 5-point smoothed line. Footprint clustering corresponds to the standard deviation from the average position of tarsal segments during stance onset. Tripod and tetrapod indexes correspond to the fraction of frames the animal display tripod or tetrapod configurations, respectively. Since many of the measured gait parameters are speed-dependent[26,29], we analysed the data for these parameters by firstly determining the best-fit regression model for the control experiment. Subsequently the residual values for each experimental group in relation to this regression model were determined, using Rstudio 1.1.442[27]. The residual data set was then tested for normality and homoscedasticity using the Shapiro-Wilk and Levene Tests. Statistical significance between the control and each of the experimental groups was determined using Kruskal-Wallis analysis of variance followed by Dunn's post hoc test (for non-normal distributions) or one-way-Anova followed by Tukey's post hoc test (for normal distributions). Boxplots represent the median as the middle line, with the lower and upper edges of the boxes representing the 25 and 75% quartiles, respectively. The whiskers represent the range of the full data set, excluding outliers. Outliers are defined as any value that is 1.5 times the interquartile range below or above the 25 and 75% quartiles, respectively. Statistical analysis was performed using custom Python scripts and GraphPad Prism.

## Statistics and reproductibility

Box and whiskers plots represent the following parameters. Boxes represent 10 to 90 percentiles and the limit of the whiskers are delimited in the minimum and maximum values. The Median is represented in the horizontal line. Box plots represent the minimum and maximum values, being the horizontal line the average representation. The performed statistical test was a two-tailed unpaired *t*-test, unless otherwise specified.

## Reporting summary

Further information on research design is available in the Nature Portfolio Reporting Summary linked to this article.

## Data availability

The RNAseq data generated and used was deposited and publicly available under accession code GSE226744. The individual GEO entries can be accessed at the following links: Control A [https://www.ncbi.nlm.nih.gov/geo/query/acc.cgi?acc=GSM7083017] Control B [https://www.ncbi.nlm.nih.gov/geo/query/acc.cgi?acc=GSM7083018] Control C [https://www.ncbi.nlm.nih.gov/geo/query/acc.cgi?acc=GSM7083019] Control CALXA [https://www.ncbi.nlm.nih.gov/geo/query/acc.cgi?acc=GSM7083020] Control CALXB [https://www.ncbi.nlm.nih.gov/geo/query/acc.cgi?acc=GSM7083021] Control CALXC [https://www.ncbi.nlm.nih.gov/geo/query/acc.cgi?acc=GSM7083022] Disease A [https://www.ncbi.nlm.nih.gov/geo/query/acc.cgi?acc=GSM7083023] Disease B [https://www.ncbi.nlm.nih.gov/geo/query/acc.cgi?acc=GSM7083024] Disease C [https://www.ncbi.nlm.nih.gov/geo/query/acc.cgi?acc=GSM7083025] Disease CALXA [https://www.ncbi.nlm.nih.gov/geo/query/acc.cgi?acc=GSM7083026] Disease CALXB [https://www.ncbi.nlm.nih.gov/geo/query/acc.cgi?acc=GSM7083027] Disease CALXC [https://www.ncbi.nlm.nih.gov/geo/query/acc.cgi?acc=GSM7083028]. The NMR data of yeast metabolites is available at https://github.com/lgafeira/Genipin-alphaSynuclein. The LC-MS data of genipin was deposited in MassIVE under the accession code MSV000091485 (doi: 10.25345/C5NS0M75B). The reporting summary for this article is available in the Supplementary Information section. All the other data supporting this study are available within this Article, Supplementary Information, and Source Data file. Yeast Growth Curves – R and R Studio plugin is fully described in https://doi.org/10.3390/antiox9090789 (https://www.mdpi.com/2076-3921/9/9/789). FlyWalker software package described in doi:10.7554/eLife.00231 (https://elifesciences.org/articles/00231). Source data are provided with this paper.

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

## Acknowledgements

We thank Carolina Conceição for the preliminary genipin toxicity assays in flies and Mariana Franco for tracking the raw FlyWalker videos. Moreover, we also acknowledge Gordon McDougall and William Allwood for the exceptional efforts in the final stages for genipin MS analysis. iNOVA4Health Research Unit (LISBOA-01-0145-FEDER-007344), which is cofunded by Fundação para a Ciência e Tecnologia (FCT) / Ministério da Ciência e do Ensino Superior, through national funds, and by FEDER under the PT2020 Partnership Agreement, is acknowledged (UIDB/04462/2020 and UIDP/04462/2020) as well LS4FUTURE Associated Laboratory (LA/P/0087/2020). Authors would like to acknowledge FCT for financial support: RRR (SFRH/BD/116597/2016) DM (2021.05505.BD); LGG (contract according to DL57/2016, [SFRH/BPD/111100/2015]); CSM (PTDC/BIA-COM/0151/2020). RM, CNS, AF and DS acknowledge funding via BacHBerry (Project No. FP7-613793; www.bachberry.eu). CNS acknowledge the European Research Council (ERC) under the European Union's Horizon 2020 Research and Innovation Programme under Grant Agreement No. 804229. RM is funded by the FCT Scientific Employment Stimulus Contract CEEC/04567/CBIOS/2020. MZ was supported by the DFG Collaborative Research Center SFB860 (project B2). TFO was supported by the DFG Center for Nanoscale Microscopy and Molecular Physiology of the Brain (CNMPB). PMD and G.M.P. were supported by grants from La Caixa Foundation (LCF/PR/HR17/52150018), FCT/Aga Khan Foundation (FCT AGA-KHAN / 541141368/2019) and by FCT, through MOSTMICRO-ITQB R&D Unit (UIDB/04612/2020, UIDP/04612/2020). The authors also thank CONGENTO: consortium for genetically tractable organisms. The NMR metabolomic data were acquired at CERMAX, ITQB-NOVA, Oeiras, Portugal with equipment funded by FCT, project AAC 01/SAICT/2016. DS and AF acknowledge support from the Scottish Government: Rural & Environment Science & Analytical Services.

## Author contributions

C.N.S. and P.M.D. conceived the study. C.N.S., R.R.R., P.M.D., and G.M.P. designed the experiments. R.R.R. and C.N.S. wrote the manuscript. D.M.S. and M.O. performed PEF fractionation, A.F. and D.S. performed LC-MS and data analysis. R.R.R. performed all the yeast experiments with some support of D.M. D.M. performed qPCR experiments and R.R.R. analysed the data. A.I.O. and M.Z. conducted protein NMR analysis. M.S. performed transmitted electron microscopy, L.G.G. performed metabolite quantification by NMR, G.M.P. performed fly experiments and C.S.M. performed Fly Walker. D.S., T.F.O., M.C.S., and R.M. contributed to the discussion of the data and revision of the manuscript. All authors read and edited the manuscript.

## Competing interests

The authors declare no competing interests.
