## [Peer Review File · Nature Communications]

REVIEWER COMMENTS

Reviewer #1 (Remarks to the Author):

The article "Genipin modulates alpha-synuclein aggregation and toxicity by promoting a metabolic shift and by modulating lipid storage" deals with the identification of the bioactive molecule from a complex mixture previously reported as protective against α Syn pathology in the context of Parkinson's Disease and to characterize its mechanism of action. The work of Rosado-Ramos et al. represents an interesting contribution to decipher the α Syn pathology.

With respect to the interdisciplinary collaboration project, my opinion on the mass spectrometry data was requested. Therefore, my comments are mainly directed to the HPLC fractionation and mass spectrometric identification of unknowns in the *C. album* leaf extract.

In general, the used HPLC and mass spectrometry methods are described not completely. Therefore, it is not possible to reproduce the data in a unambiguous way, and I recommend major revisions of this part. The reasons for this decision are summarized in the following:

1. line 128 ("Importantly, when incubated with genipin, the percentage of yeast cells displaying these inclusions decreased to 59.6% (Figure 2B, right panel).")

The specified value with one decimal place is questionable to my opinion. What was the number of replicates and how much was the deviation?

2. line 165: The authors used a 100-fold excess of genipin. For me, it is not clear why such a high concentration was applied.

3. line 490ff (description of preparative HPLC): The following data is missing:

- HPLC instrumentation
- column dimensions
- injection volume
- method for solvent evaporation of collected fractions

4. Chapter "Phenolic profile determination by LC-MS":

- line 526: typo? 3.5 kV is more likely than 23.5 kV. The same for tube lens voltage: 210 V instead of 2100 V

- line 536: Software for feature extraction etc. is not defined? Overall, the procedure for identification of unknowns is not comprehensively described.

5. Supplementary Figure 1 and Table 1:

“Peaks present in these fractions were tentatively annotated by utilizing accurate mass (<3ppm), MS/MS fragmentation patterns and reference to retention time for previously analyzed standards (Supplementary table 1).”

- Please highlight potential iridoids in Table 1 for better visibility

- Only 2 compounds were identified with level 1 (ferulic acid and (-)-epicatechin). That is a very low number in comparison to the list of putatively identified peaks. Please comment.

- My main concern is that the target compound genipin was not verified with the available authentic standard, albeit it is available and used by the authors for further experiments. In Supp. Table 1 genipin is only Level 3.

6. Line 662: Please add the respective reference for the extraction method.

Reviewer #2 (Remarks to the Author):

The work presented by the authors is remarkable for its findings, number and types of essays, robustness and depth of the results. The authors show the continuation of a previously published study, where the impact of the polyphenol-enriched fraction of the crude extract of *Corema album* on the toxicity and aggregation of α Syn, key in Parkinson's disease, was evaluated. On this occasion, through a bioguided fractionation, they identify the iridoid genipin as a possible compound responsible for the effect. The study is completed with tests aimed at determining the mechanism by which it exerts its biological effect.

At this point it should be noted that the authors specify carrying out a bioguided study or bioguided fractionation of fractions 1 and 6 of the PEF, although there are many aspects not addressed in this regard. On the one hand, where and who collects and identifies the species (herbarium code of the corresponding institution). Aspects such as the starting mass and fractions, together with the yield of the related compound/derivatives to which the biological effect is attributed. It is not specified whether the genipin compound used for the tests comes from plant material or was purchased commercially. Even in Table 1 (Supplementary Information), genipin is listed as a potential in-source fragment of an iridoid derivative. It is not specified whether, as in the case of other plant species, a precursor such as geniposide or only the ester is present. In any case, its characterization (HRMS, 1-2 D NMR, chemical

shifts, although not described here for the first time) should be included in the supplementary information.

On the other hand, if it is pointed out that due to the presence of the representative iridoid genipin, the activity of the fractions is present, this must be justified with the preliminary biological activity of the rest of the compounds found, together with the proportion in which they are found in the mixture, after being isolated. Another noteworthy fact is that genipin and related compounds are only isolated from F1. It is well known that many extracts or mixed fractions of natural origin have antagonistic, synergistic or additive effects, and on some occasions the activity they present is not due to the main or major compounds but to the mixture. Here, it is necessary to clarify and expand the information on this point, to validate the proposed hypothesis. Otherwise, the axis of biodirected fractionation can only support the protocol carried out on the genipin compound in particular, supported by bibliographic data from previous reports, which does not detract from its validity, it only changes the starting point of the extensive work performed.

I consider that, if the aforementioned aspects are addressed in the review, the manuscript is a highly significant piece of work for this area of research.

Minor considerations:

The figures, diagrams and tables are clear and summarize the information to be shown. There are some minor details, like:

- The low quality of figure 4.
- The length of figure legends (in many cases, experimental procedures are detailed that should be in materials and methods, directing the reader there).
- The dotted and dashed lines in figure 1 should be of different colors to improve visualization.
- Table 1 (supplementary information) should be divided into 2 tables according to F1 and F6. In addition, the chemical formulas must have the atomicity as a subscript, and it would improve the visualization if there are borders in it.
- In Figure 3, the data that is written in the HSQC spectrum is not displayed well (put it in bold or just leave it as a caption).
- Correct to justified format of the document.
- Define β Syn-GFP when it is named for the first time and check that all abbreviations are defined (LBs, FM4-64, SNCA, etc)

The work is very well written and discussed, so that it is very clear when it comes to being approached in its reading. A minor detail is that some abbreviations are in a different format, check uppercase and lowercase letters. Some sentences could be clarified:

- - Page 4, line 56-58. What happens to the excess interactions in membranes?

- - Page 4, line 63. In the "present" study has been carried out..
- - Page 4, line 81. Send the reader to the experimental part to consult the fractionation conditions.
- - Page 7, line 159. Not understood: ... we "resolved" α Syn species by SEC and monomeric...
- - Page 8, line 164, 175-176. Add what is observed or clarify to the reader what is denoted by the change in intensity of the NMR signal after the perturbation (briefly).
- - Page 9, line 189, complete the sentence, about what the gene condition caused by the overexpression of α Syn-GFP translates into.
- - Page 9, line 202, add some other GO detail.
- - Page 13, line 323, clarify the sentence.
- - Page 15, line 351, do references 1 and 2 correspond?
- - Page 26, line 657, Place name of authors and citation in normal format.

The referenced bibliography is correct and up-to-date. These works could be added where they review the therapeutic potential of genipin in the CNS: Yanwei, Li, et.al. [2016 CNS Drugs. DOI 10.1007/s40263-016-0369-9], and [Brys, M., Int J. Mol Sci, 2022, 23, 902]. In addition, a review could be added in the introduction where updated information about Parkinson's disease is reviewed.

Reviewer #3 (Remarks to the Author):

In this manuscript, the authors identify genipin as an inhibitor of alpha-synuclein toxicity in yeast and fly models for PD. The authors build on their previous findings that a polyphenol-enriched fraction of *Corema album* leaf extract reduces alpha-synuclein toxicity and restores autophagy in yeast. Here, the authors now fractionate this polyphenol-enriched fraction into 16 fractions and identify iridoids as bioactive compounds, among them genipin. They use genipin as representative iridoid for further analysis of its cytoprotective function and show that genipin reduces Syn aggregation in yeast and in vitro. Guided by RNA-Seq analysis, the authors present some data that link genipin effects to carbohydrate metabolism, lipid metabolism, endocytosis as well as mitochondrial functionality. However, this analysis remains rather superficial, and any cellular targets or molecular pathways causally involved in genipin cytoprotection remain rather unexplored. Neuroprotective functions of genipin have already been suggested e.g. in iPSC derived from early-onset PD patient (PMID:26797011), in a mouse model of brain inflammation (PMID:20123040), or via reduction of PolyQ aggregation and oxidative stress (PMID:24486383). Thus, it is indeed highly interesting that genipin reduces Syn

aggregation, but some more mechanistic insights into its mode of action would be beneficial to provide sufficient scientific novelty.

Some of the results presented are interesting and novel, and the finding that genipin prevents Syn aggregation in vitro as well as in yeast is of importance. However, some of the conclusions drawn by the authors are not sufficiently supported by data, e.g. that genipin provides cytoprotection by “promoting a metabolic shift” and by “modulating lipid storage”, as postulated by the authors in the title. In addition, some experiments lack controls, as detailed below.

1. In Figure 2, the authors show that Syn kills about 5% of yeast cells after 6 h of expression. Genipin treatment decreases the percentage of cell death to 1%, suggesting an almost complete inhibition of Syn-mediated cell killing. This is of course highly interesting, but the percentage of cell death is still quite low. Is the cytoprotective effect of genipin still visible upon prolonged Syn expression and thus increased rates of cell death?

2. The authors present some data showing that Syn clearance is unaffected by the presence of genipin. How does this fit with the previous analysis, where the authors used the polyphenol-enriched fraction to show that this supports Syn clearance and induces autophagy?

3. The authors performed RNA-seq of cells with/without Syn expression and treated/not treated with genipin. The conclusion drawn from this analysis is that genipin protection involves carbohydrate and lipid metabolism, endocytosis as well as protein degradation and trafficking pathways. This data is used to direct further analysis, but mostly these analyses remain rather superficial. Carbohydrate metabolism processes as given in Fig. 4B seem to be “Fructose metabolic processes” and “Maltose metabolic processes” and “Fructose transmembrane transport”. What do the authors conclude from differential expression of HXT10, GUP2, JEN1, FBP26 etc and the altered levels of lactate, glycerol, acetate as well as TCA cycle intermediates?

These findings are stated, but not put into context, and no follow-up experiments have been performed. e.g. levels of ornithine, glutamate, succinate, citrate, lactate, acetate and glycerol are measured, some of them affected by only Syn but not genipin, some of them affected by genipin but not Syn, some of them affected by both (e.g. genipin reversed the Syn effect). Very unclear what these data actually suggest?

4. The statement “Genipin decreases oxidative stress by modulating carbon metabolism” is not supported by data. The authors show an increase of ROS upon Syn expression (as shown frequently also by others) and a decrease of ROS accumulation upon genipin treatment. This does not allow a conclusion as to genipin reduces “oxidative stress by modulating carbon metabolism”

5. Also the data presented in respect to endocytosis is rather limited. The authors present qRT-PCR data for the three hits that have been found deregulated in the RNA-seq data and show:

-YPT1 upregulated upon Syn expression, and downregulated again upon genipin treatment

-HRR25 upregulated upon Syn expression and even further upregulated upon genipin treatment

-YPK1 unaffected by Syn expression, but upon additional genipin treatment upregulated.

To me, it remains unclear what these data suggest. In addition, the authors perform two different vacuolar stainings (FM4-64 and CMAC), that together imply that there is a problem with endocytosis upon Syn expression that can be restored by genipin treatment. Is there any link between this observation and the three genes that are found to be differentially expressed? How do these genes contribute to the effects of genipin on endocytosis?

6. Similarly, also the data presented in respect to lipid homeostasis is rather limited and correlative and does not provide substantial new insights or a causal connection of any of these processes to Syn toxicity and genipin cytoprotection.

The conclusion that genipin alters the interaction of Syn with the plasma membrane is unclear to me. Fig. 6E shows that Syn-GFP decorates the plasma membrane to a similar extent in untreated and genipin treated cells, though nucleation of inclusions at the membrane was reduced.

In line with previous publications, the authors find that that Syn increases neutral lipid content, which is further exacerbated by genipin treatment. A protein involved in lipid droplet biogenesis (Ldb16; mRNA found upregulated in RNA seq) was essential for genipin-induced cytoprotection (ldb16 deletion mutants still showed 5% cell death upon Syn expression, with and without addition of genipin), suggesting a link between genipin cytoprotection and lipid droplet biogenesis. Comparing microscopic analysis of Syn-GFP in wild type (presented in Fig. 2B) and in ldb16 mutants (presented in Fig. 6K), it seems that loss of ldb16 in general prominently induces the aggregation of Syn. This needs to be

assessed and directly compared in wt and ldb16 mutants using same setting. It seems that loss of ldb16 per se massively induces the aggregation of Syn (e.g numerous inclusions per cell), thus it is a bit unclear what to conclude here.

6. In general, the analysis of genipin effects in the Drosophila PD model seem a bit limited. While it is of course not necessary to confirm all findings obtained in the yeast system also in the fly model, an expression control for Syn seems necessary. In addition, it would be beneficial to at least confirm the main effect of genipin (inhibition of Syn aggregation) also in this model system (e.g. via immunoblotting as has been done in the yeast system).

In addition, the presented data needs to be improved and necessary controls need to be included in the figure panels:

- The pro-survival effect of genipin on Syn-expressing flies shown in Fig. 7A is very minor. These curves reflect the survival of only 30 flies per genotype, kept in 3 vials of 10 flies each. This needs to be repeated in independent experiments to confirm this effect, as the number of flies is very low and the effect small. In addition, control flies treated with genipin need to be included in the same experiments (in particular as genipin seems to affect control flies in a non-concentration dependent manner according to Supplemental Fig. 6 where some concentrations of genipin seemed to initially kill the control flies, some seemed to slightly rescue). Thus, all four groups need to be compared directly and statistics need be adjusted to compare these four groups.

In general, control flies treated with genipin should be included in the analyses in Figure 7. For Figure 7B, a two-way ANOVA needs to be performed (just as done by the authors for Supplemental Figure 6B, showing corresponding climbing activity in the control flies.) Moreover, control flies treated with the same concentration of genipin should be shown in Fig. 7B.

In respect to the footprint clustering as well as tripod and tetrapod index: it is a bit unclear how this adds additional value beyond what is already shown in the climbing activity in 7B.

Is there any reason to use transgenic flies expressing Syn-GFP for survival and climbing activity and flies expressing untagged Syn for this extended analysis of motor defects?

The quantification of the footprint analysis shown in Fig. 7C is not convincing. There is no significant effect between Syn and Syn+genipin. Thus, the conclusion here would be that genipin has no significant rescuing effect.

Minor:

Line 56-58: The following statement should be supported by references:

“However, in pathological conditions, alpha-Syn is described to exert toxic effects, not only by its aggregation process, but also by performing excessive membrane interactions”

Line 131: Please correct, the sentence is unclear

“..., yeast cells tend to form less detergent-insoluble with SDS or a greater portion of smaller inclusions”

Line 111: The authors do not find any effect of genipin on growth arrest induced by FUS and Htt expression in yeast and thus state “The lack of effect of genipin on cell growth in these models reinforces the potential of genipin to be a specific bioactive molecule to counteract alphaSyn-GFP toxicity and aggregation”. As genipin has been shown by others to be neuroprotective in different model systems and scenarios, this statement should be corrected.

Reviewer #4 (Remarks to the Author):

The paper focuses on the investigation of a mixture of natural leaf extracts from *Corema album*, previously shown to have protective effects against alpha-synuclein pathology. Genipin is identified as the major responsible for this effect and several complementary techniques are used to demonstrate and understand the origins of this effect. The topic is certainly interesting and it builds on previous work by the PI.

The data convincingly show a reduction in the formation of alpha-synuclein aggregates in presence of genipin. The authors report insights achieved through NMR on different parts of the protein in a residue resolved manner. Interestingly different effects are monitored along the primary sequence ranging from

intensity changes (N-ter), to essentially no effect (C-ter) to a combination of intensity changes/chemical shift perturbations (60-80, 100-120) residues. It would be interesting to discuss these findings more in detail also in terms of possible structural and dynamic variations induced by genipin and compare them to effects of other compounds that are known to interact with alpha-synuclein (EGCG for example).

The authors also discuss the link between alpha-synuclein and lipid metabolism (paragraph 7). Would it be possible to access residue resolved information on this topic? For example by investigating by NMR the properties of alpha-synuclein in presence of lipids, or of membrane models, with and without genipin?

Reviewer #5 (Remarks to the Author):

The manuscript is dedicated to the study of anti-aggregation properties of the Genipin substance from *Corema album*. The author have shown that genipin affects the alpha-synuclein aggregation and toxicity. Different plants are actively used as a source of anti-amyloid agents. The author have found not only that the leaves contain some substance, but also identified it, which allowed more detailed study of its effects. The genipin has been shown non-toxic for yeast and fruit flies, and prevent the effects of alfa-synuclein overproduction. Though the results provided in the manuscript are very interesting, there are some comments:

1. The yeast-based model are actively used for study of alfa-synuclein aggregation. Though it is an adequate model for the study, a lot of molecular systems in yeast cells are quite different. The same with fruit flies. Have the author tested the effects of genipin in mammal-based models?
2. The same question with the metabolic changes in yeast cells. The molecular systems might be quite different, so it would be better to show similar results with transcriptomic analysis in mammals.
3. Though the authors have precisely described the effects of genipin, no molecular mechanisms explaining the effects are provided. Which proteins does genipin interact with? How does it affect their activity?
4. The bioinformatic analysis should be described in more detailes. The important steps are missing.
5. There are no data how the synuclein and genipin affects the neural tissue of flies, only results of physiological tests are provided.
6. If genipin directly prevents alfa-Syn aggregation by interaction with monomers, it would be better to confirm this interaction by providing its model and mutants unable to interact with genipin.

REVIEWER COMMENTS

Reviewer #1 (Remarks to the Author):

The article "Genipin modulates alpha-synuclein aggregation and toxicity by promoting a metabolic shift and by modulating lipid storage" deals with the identification of the bioactive molecule from a complex mixture previously reported as protective against α Syn pathology in the context of Parkinson's Disease and to characterize its mechanism of action. The work of Rosado-Ramos et al. represents an interesting contribution to decipher the α Syn pathology.

With respect to the interdisciplinary collaboration project, my opinion on the mass spectrometry data was requested. Therefore, my comments are mainly directed to the HPLC fractionation and mass spectrometric identification of unknowns in the C. album leaf extract.

In general, the used HPLC and mass spectrometry methods are described not completely. Therefore, it is not possible to reproduce the data in an unambiguous way, and I recommend major revisions of this part. The reasons for this decision are summarized in the following:

1. line 128 ("Importantly, when incubated with genipin, the percentage of yeast cells displaying these inclusions decreased to 59.6% (Figure 2B, right panel)."

The specified value with one decimal place is questionable to my opinion. What was the number of replicates and how much was the deviation?

Indeed, the decimal place is not necessary, it was changed in the text and deviation was added (60 ± 6). The graph in figure 2B was generated using data from 4 independent experiments and the deviation was (in the case of aSyn overexpressing cells treated with genipin).

2. line 165: The authors used a 100-fold excess of genipin. For me, it is not clear why such a high concentration was applied.

Indeed, the concentration used in NMR experiment is higher than in every other experiment. It is important to note that protein NMR is not a very sensitive method, so we need to increase concentrations in order to observe an effect visible in the equipment. Most importantly, genipin at a much lower concentration (10 μ M) in the same *in vitro* configuration is able to almost abolish aSyn aggregation, as shown in Figure 3A, B, C and E.

3. line 490ff (description of preparative HPLC): The following data is missing:

- HPLC instrumentation
- column dimensions

- injection volume
- method for solvent evaporation of collected fractions

The information was added as suggested.

4. Chapter "Phenolic profile determination by LC-MS":

- line 526: typo? 3.5 kV is more likely than 23.5 kV. The same for tube lens voltage: 210 V instead of 2100 V

The voltage was corrected, the "2" was indeed a typo.

- line 536: Software for feature extraction etc. is not defined? Overall, the procedure for identification of unknowns is not comprehensively described.

The description of the method used to identify the unknowns was improved and the software used was specified.

5. Supplementary Figure 1 and Table 1:

"Peaks present in these fractions were tentatively annotated by utilizing accurate mass (<3ppm), MS/MS fragmentation patterns and reference to retention time for previously analyzed standards (Supplementary table 1)."

- Please highlight potential iridoids in Table 1 for better visibility

Potential iridoids were highlighted in blue.

- Only 2 compounds were identified with level 1 (ferulic acid and (-)-epicatechin). That is a very low number in comparison to the list of putatively identified peaks. Please comment.

We agree that one of the limitations of LC-MS analysis in complex plant metabolites mixtures is to have commercially available authentic chemical standard to be analysed under same analytical conditions and allowing more level 1 annotations. However, many of the compounds present in the fractions are not commercially available and therefore we did not have them in our in-house library. The putative identification of genipin derivatives in the bioactive fractions was supported by the confirmation of bioactivity using commercial genipin aglycone. This suggests that the original bioactive agents could have been the genipin components.

- My main concern is that the target compound genipin was not verified with the available authentic standard, albeit it is available and used by the authors for further experiments. In Supp. Table 1 genipin is only Level 3.

Regarding the validation of genipin: it has become clear from our LC-MS studies with the authentic genipin standard (see figures 1 to 4 below) that the m/z signal recorded in fraction 1 (that was matched with genipin) were in-source fragments suggesting the presence of component with genipin as the aglycone. Therefore, we cannot match these peaks with genipin itself but can suggest the presence of genipin components. This use of genipin as a representative compound of the iridoid class was highlighted and clarified.

Figure 1. MS properties of genipin

Full scan MS chromatograph of genipin at 5 ppm in positive mode and MS spectra under the only MS peak at 14.94 min. Main m/z value = 209.0802 = $[M-H_2O]^+$ ion, gives expected predicted formula of $C_{11}H_{13}O_4$ at < 1 ppm. Molecular ion = 227.0907 = $[M+H]^+$ ion, gives expected predicted formula of $C_{11}H_{15}O_5$ at < 1 ppm.

Figure 2. MS² properties of genipin (+ve mode)

MS² spectra show shared fragments from the m/z 209.08 $[M-H_2O]^+$ ion and the m/z 227.09 molecular ion $[M+H]^+$ with predictable losses.

Genipin 5ppm Neg Full MS

Figure 3. MS properties of genipin (- mode)

Full scan MS chromatograph of genipin at 5 ppm in negative mode and MS spectra under the main MS peak at 14.89 min. The m/z value of 225.0762 gives a predicted formula of $C_{11}H_{12}O_5$.

Figure 4. MS² properties of genipin (- mode)

MS² spectra of the *m/z* 225.08 molecular ion [M-H]⁻ shows predictable losses and is consistent with previous data (PubChem 442424).

The contaminant present in the standard (RT 12.92) has a *m/z* of 433.10 and yields MS² spectra consistent with a formic acid adduct of a glycoside of genipin; i.e., neutral losses of 46 (formic acid) to give *m/z* 387 and loss of 208 (formic acid + glucosyl) to give *m/z* 225. The *m/z* value 433.1340 yields a predicted formula C₁₈H₂₅O₁₂; ppm < 1.

6. Line 662: Please add the respective reference for the extraction method.

The reference was added as requested.

Reviewer #2 (Remarks to the Author):

The work presented by the authors is remarkable for its findings, number and types of essays, robustness and depth of the results. The authors show the continuation of a previously published study, where the impact of the polyphenol-enriched fraction of the crude extract of *Corema album* on the toxicity and aggregation of α Syn, key in Parkinson's disease, was evaluated. On this occasion, through a biodirected fractionation, they identify the iridoid genipin as a possible compound responsible for the effect. The study is completed with tests aimed at determining the mechanism by which it exerts its biological effect.

At this point it should be noted that the authors specify carrying out a bioguided study or bioguided fractionation of fractions 1 and 6 of the PEF, although there are many aspects not addressed in this regard. On the one hand, where and who collects and identifies the species (herbarium code of the corresponding institution).

Herbarium code and institution was added, as suggested.

Aspects such as the starting mass and fractions, together with the yield of the related compound/derivatives to which the biological effect is attributed. It is not specified whether the genipin compound used for the tests comes from plant material or was purchased commercially. Even in Table 1 (Supplementary Information), genipin is listed as a potential in-source fragment of an iridoid derivative. It is not specified whether, as in the case of other plant species, a precursor such as geniposide or only the ester is present. In any case, its characterization (HRMS, 1-2 D NMR, chemical shifts, although not described here for the first time) should be included in the supplementary information. On the other hand, if it is pointed out that due to the presence of the representative iridoid genipin, the activity of the fractions is present, this must be justified with the preliminary biological activity of the rest of the compounds found, together with the proportion in which they are found in the mixture, after being isolated.

We appreciate this comment and in fact our text was not clear and led to the reviewer asking for clarification. Considering the widespread presence of iridoid derivatives in the bioactive fractions, we hypothesised that this class of compounds could be associated with the bioactivity observed because of their presence only in the bioactive fractions. Therefore we decided to test the commercially available aglycone of these derivatives, genipin, and used pure commercially available genipin as a representative iridoid core structure in the bioactivity studies. Moreover this selection of the aglycone for further bioactivity analysis is also related to the fact that this is the core structure that will survive gut metabolism before phase II reactions like sulfation. The glucosides are hydrolysed in the gut, and therefore the aglycone will have much more physiological relevance and importance to a future clinic application. The fact that we retained the bioactivity for genipin that increases cell growth in yeast overexpressing \$\alpha\$ Syn lead us to proceed with this commercial compound.

The fractionation method does not refer to an isolation method. It was used as tool to search for possible bioactive molecules therefore we did not use fractions from plant material to study bioactivity and that is the reason we did not calculate yield or further chemical characterization of "isolated" compounds. In fact the genipin used in bioactivity assays have accordingly to the provider a purity (HPLC) of min 98% and that identity conforms to the structure by ¹H NMR, but also MS and MS² data (supplied if required, see figure 1 -4 in answer to review 1).

We added information to clarify the origin and the selection of genipin in the presented study.

Another noteworthy fact is that genipin and related compounds are only isolated from F1. It is well known that many extracts or mixed fractions of natural origin have antagonistic, synergistic or additive effects, and on some occasions the activity they present is not due to the main or major compounds but to the mixture. Here, it is necessary to clarify and expand the information on this point, to validate the proposed hypothesis. Otherwise, the axis of biodirected fractionation can only support the protocol carried out on the genipin compound in particular, supported by bibliographic data from previous reports, which does not detract from its validity, it only changes the starting point of the extensive work performed.

We agree that in many plant extracts the present compounds have antagonist, synergistic or additive effects and that the bioactivity of a mixture could be due to these effects and not an isolated compound. As stated in the previous answer our strategy was to test the aglycone of several derivatives present in the mixture (F1 and F6, both containing these genipin derivatives and presenting bioactivity). Fortunately, we detected a very impressive activity in improving the growth of aSyn expressing yeast with commercial genipin. This fact led us to continue with this compound in the following experiments. The referee is correct that the starting point of the work could be to test only genipin, but in reality our rational started in a previous publication¹ where we have an impressive activity of a (poly)phenol enriched fraction from *Corema album* leaves and we also tested for the bioactivity the major compounds present in the extract without success. This work was the reason for performing the bioactivity guided fractionation in the current manuscript. We consider that this is an important information that should be highlighted as it shows how starting from a raw material and after testing the major compounds isolated we can only proceed after a bioactivity bioguided fractionation to elucidate possible bioactive compounds.

Another important aspect in the analysis of the components in the bioactive fractions was to identify what where common structures. After identifying genipin derivatives, mainly glucosides, we studied what is described about the mammalian metabolism of these compounds² and concluded that we should test the aglycone for further bioactivity since it will have much more physiological relevance as explained before.

This rational was clarified in text to facilitate the understanding of genipin selection.

Minor considerations:

The figures, diagrams and tables are clear and summarize the information to be shown.

There are some minor details, like:

- The low quality of figure 4.

The quality of the figure was improved as requested

- The length of figure legends (in many cases, experimental procedures are detailed that should be in materials and methods, directing the reader there).

legends were revised to remove methodological details as requested

- The dotted and dashed lines in figure 1 should be of different colors to improve visualization.

The lines are now in different colours, as suggested.

- Table 1 (supplementary information) should be divided into 2 tables according to F1 and F6. In addition, the chemical formulas must have the atomicity as a subscript, and it would improve the visualization if there are borders in it.

Table 1 was divided in two and formatted as suggested.

- In Figure 3, the data that is written in the HSQC spectrum is not displayed well (put it in bold or just leave it as a caption).

The data written in the HSQC was improved as suggested and placed aside to the spectrum.

- Correct to justified format of the document.

We acknowledge the suggestion, however, the instructions from Nature Communications are clear and the following: "The manuscript file should be formatted as double-spaced, single-column text without justification."

- Define α Syn-GFP when it is named for the first time and check that all abbreviations are defined (LBs, FM4-64, SNCA, etc)

All abbreviations checked as requested.

The work is very well written and discussed, so that it is very clear when it comes to being approached in its reading. A minor detail is that some abbreviations are in a different format, check uppercase and lowercase letters.

All the abbreviations were uniformized.

Some sentences could be clarified (all checked):

- - Page 4, line 56-58. What happens to the excess interactions in membranes?

The sentence was rewritten to make it clearer

- - Page 4, line 63. In the "present" study has been carried out..

- - Page 4, line 81. Send the reader to the experimental part to consult the fractionation conditions.

- - Page 7, line 159. Not understood: ... we "resolved" α Syn species by SEC and monomeric...

The sentences were corrected as suggested

- - Page 8, line 164, 175-176. Add what is observed or clarify to the reader what is denoted by the change in intensity of the NMR signal after the perturbation (briefly).

A brief description was added to help the reader to interpret the change in intensity of the NMR signal as requested

- - Page 9, line 189, complete the sentence, about what the gene condition caused by the overexpression of α Syn-GFP translates into.

- - Page 9, line 202, add some other GO detail.

- - Page 13, line 323, clarify the sentence.

Sentences were completed and clarified as requested

- - Page 15, line 351, do references 1 and 2 correspond?

Reference 1 and 2 do not correspond. The reference manager was refreshed and the correct number of those two references was assigned.

- - Page 26, line 657, Place name of authors and citation in normal format.

The correct formatting of the references was added.

The referenced bibliography is correct and up-to-date. These works could be added where they review the therapeutic potential of genipin in the CNS: Yanwei, Li, et.al. [2016 CNS Drugs. DOI 10.1007/s40263-016-0369-9], and [Brys, M., Int J. Mol Sci, 2022,

23, 902]. In addition, a review could be added in the introduction where updated information about Parkinson's disease is reviewed.

The references where added as suggested.

Bibliography

1. Macedo, D. *et al.* (Poly)phenols protect from α -synuclein toxicity by reducing oxidative stress and promoting autophagy. *Human Molecular Genetics* **24**, 1717–1732 (2015).
2. Hou, Y. C., Tsai, S. Y., Lai, P. Y., Chen, Y. S. & Chao, P. D. L. Metabolism and pharmacokinetics of genipin and geniposide in rats. *Food and Chemical Toxicology* **46**, 2764–2769 (2008).

Reviewer #3 (Remarks to the Author):

In this manuscript, the authors identify genipin as an inhibitor of alpha-synuclein toxicity in yeast and fly models for PD. The authors build on their previous findings that a polyphenol-enriched fraction of *Corema album* leaf extract reduces alpha-synuclein toxicity and restores autophagy in yeast. Here, the authors now fractionate this polyphenol-enriched fraction into 16 fractions and identify iridoids as bioactive compounds, among them genipin. They use genipin as representative iridoid for further analysis of its cytoprotective function and show that genipin reduces Syn aggregation in yeast and *in vitro*. Guided by RNA-Seq analysis, the authors present some data that link genipin effects to carbohydrate metabolism, lipid metabolism, endocytosis as well as mitochondrial functionality. However, this analysis remains rather superficial, and any cellular targets or molecular pathways causally involved in genipin cytoprotection remain rather unexplored. Neuroprotective functions of genipin have already been suggested e.g. in iPSC derived from early-onset PD patient (PMID:26797011), in a mouse model of brain inflammation (PMID:20123040), or via reduction of PolyQ aggregation and oxidative stress (PMID:24486383). Thus, it is indeed highly interesting that genipin reduces Syn aggregation, but some more mechanistic insights into its mode of action would be beneficial to provide sufficient scientific novelty. Some of the results presented are interesting and novel, and the finding that genipin prevents Syn aggregation *in vitro* as well as in yeast is of importance. However, some of the conclusions drawn by the authors are not sufficiently supported by data, e.g. that genipin provides cytoprotection by "promoting a metabolic shift" and by "modulating lipid storage", as postulated by the authors in the title.

The reviewer is right in fact genipin have been described with several activities that is normally described for polyphenols due to their pleiotropic activities. However our work is the first describing its effect in reducing Syn aggregation *in vitro and in vivo* (yeast and now also included new data for drosophila, Figure 7D). We introduced the references of the other activities as indicated by the reviewer and improved the discussion on our data concerning genipin mechanism. Of notice that we believe that the most significant contribution of this work is the fact of genipin interfering with syn aggregagtion and for that we explore *in vitro* assays (Tht, DLS, EM, NMR), *in vivo* mechanism in yeast (RNAseq, metabolites by NMR, gene expression by RT-PCR and use of mutants to understand dependent mechanisms) and finally we validate the beneficial effect of aSyn in a drosophila model of PD that also aggregates aSyn and present locomotor deficits. We also have improved the mechanistic studies by including one more yeast relevant mutant (Figure 6 O-Q) and confirm reduction of aSyn insoluble forms in the brain of drosophila (Figure 7D). The title was changed to reflect better our findings.

1. In Figure 2, the authors show that Syn kills about 5% of yeast cells after 6 h of expression. Genipin treatment decreases the percentage of cell death to 1%, suggesting

an almost complete inhibition of Syn-mediated cell killing. This is of course highly interesting, but the percentage of cell death is still quite low. Is the cytoprotective effect of genipin still visible upon prolonged Syn expression and thus increased rates of cell death?

In fact, in figure 2 the observed improvement of cell death by 10 μ M of genipin is monitored at 6h of expression (cells in exponential phase). In Figure 2A-E we intend to follow the effect of genipin in aSyn early stages of aggregates formation and consequently we measure cell viability at the same time point. Moreover the setup of 6 h is to get the window for evaluation of effects in number of cells with inclusions. Prolonged times will not allow an accurate measure of cells with inclusions due to the fact that cells are overloaded of inclusions. This lag between aSyn induction and cell death was already described for the model as the opportunity to study the protein's biological effects before toxicity confounded results. The steady state levels and distribution of aSyn is achieved at 8h of expression³. Therefore the reviewer is right, by inducing the aSyn expression in cells in exponential phase for prolonged times will conduct to increased cell death and in this setup we confirmed that after 24h of aSyn-GFP expression we cannot detect a protection with genipin. However in Figure 1C we evaluate cells growth also in the presence of 10 μ M of genipin since the beginning (cells in lag phase) and we clearly see an improvement until 24 hours indicating the efficacy of genipin, in this different experimental setup. In the end, the most striking observation and that reinforces the activity and potential of genipin to reduce toxicity due to aSyn expression is the *in vivo* validation in drosophila (Figure 7A).

2. The authors present some data showing that Syn clearance is unaffected by the presence of genipin. How does this fit with the previous analysis, where the authors used the polyphenol-enriched fraction to show that this supports Syn clearance and induces autophagy?

The reviewer is correct. In fact, we previously show that the polyphenol-enriched fraction (PEF) of *Corema album* leaves, is capable to improve aSyn-GFP clearance by improving autophagic flux. However, PEF is a very complex mixture of compounds and we cannot assume that this specific effect is a result of the genipin derivatives. Thus, the effect observed can be undertaken by any other compound or by a synergy of different compounds, and cannot be directly compared with the bioactivity of genipin in this work. This aspect was clarified in discussion.

3. The authors performed RNA-seq of cells with/without Syn expression and treated/not treated with genipin. The conclusion drawn from this analysis is that genipin protection involves carbohydrate and lipid metabolism, endocytosis as well as protein degradation and trafficking pathways. This data is used to direct further analysis, but mostly these analyses remain rather superficial. Carbohydrate metabolism

processes as given in Fig. 4B seem to be “Fructose metabolic processes” and “Maltose metabolic processes” and “Fructose transmembrane transport”. What do the authors conclude from differential expression of HXT10, GUP2, JEN1, FBP26 etc and the altered levels of lactate, glycerol, acetate as well as TCA cycle intermediates?

These findings are stated, but not put into context, and no follow-up experiments have been performed. e.g. levels of ornithine, glutamate, succinate, citrate, lactate, acetate and glycerol are measured, some of them affected by only Syn but not genipin, some of them affected by genipin but not Syn, some of them affected by both (e.g. genipin reversed the Syn effect). Very unclear what these data actually suggest?

Indeed, regarding the effects of aSyn-GFP overexpression and genipin protection in yeast metabolism, our data is very descriptive. As we could conclude from our study, genipin presents a marked pleiotropic effect (as represented in Figure 4).

Given the robustness of all the metabolic processes, the use of, for example, deletion mutants turns to be ineffective once there are compensatory mechanisms performed by enzymes' isoforms. Also, the measure of metabolites, without any further evaluation, as fluxes, is rather limited. To further dissect the target enzymes or pathways involved in this action, a different experimental approach, as the use of 'metabolic stories', alone, or combined with a more in depth transcriptomic approach, must be used.

So, as we are aware of all these limitations, we kept a descriptive approach regarding the effect of genipin protection in yeast cells overexpressing aSyn-GFP and focused on describe all the important activities of genipin in other important cellular hallmarks of PD.

Nevertheless, we improved the discussion on this topic as suggested.

4. The statement “Genipin decreases oxidative stress by modulating carbon metabolism” is not supported by data. The authors show an increase of ROS upon Syn expression (as shown frequently also by others) and a decrease of ROS accumulation upon genipin treatment. This does not allow a conclusion as to genipin reduces “oxidative stress by modulating carbon metabolism”

We agree that we cannot claim that and in that sense the sentence was clarified.

5. Also the data presented in respect to endocytosis is rather limited. The authors present qRT-PCR data for the three hits that have been found deregulated in the RNA-seq data and show:

-YPT1 upregulated upon Syn expression, and downregulated again upon genipin treatment

-HRR25 upregulated upon Syn expression and even further upregulated upon genipin treatment

-YPK1 unaffected by Syn expression, but upon additional genipin treatment upregulated.

To me, it remains unclear what these data suggest. In addition, the authors perform two different vacuolar stainings (FM4-64 and CMAC), that together imply that there is a problem with endocytosis upon Syn expression that can be restored by genipin treatment. Is there any link between this observation and the three genes that are found to be differentially expressed? How do these genes contribute to the effects of genipin on endocytosis?

The discussion was improved to relate the genes alterations to the observation of genipin's improvement in aSyn-impaired endocytosis, as suggested.

6. Similarly, also the data presented in respect to lipid homeostasis is rather limited and correlative and does not provide substantial new insights or a causal connection of any of these processes to Syn toxicity and genipin cytoprotection.

To improve the connection between lipid homeostasis with syn toxicity and genipin cytoprotection we decided to perform a new experiment using Erg6 deletion mutant. Erg6 converts zymosterol to fecosterol in the ergosterol biosynthetic pathway and *erg6* mutant accumulates zymosterol but cannot synthesize ergosterol. We observe that this mutant is able to form aSyn aggregates and that genipin can still reduce the number of cells with inclusions (new figure 6 O-P) reinforcing that the protective affect of genipin should be focused in an improvement of endocytosis rather than in impacting the plasma membrane composition. We include the new data and discuss this aspect.

The conclusion that genipin alters the interaction of Syn with the plasma membrane is unclear to me. Fig. 6E shows that Syn-GFP decorates the plasma membrane to a similar extend in untreated and genipin treated cells, though nucleation of inclusions at the membrane was reduced.

Sentence was re-written to clarify that what we observe was decrease in nucleation of inclusions in the membrane and not a reduction in aSyn-membrane interaction.

In line with previous publications, the authors find that that Syn increases neutral lipid content, which is further exacerbated by genipin treatment. A protein involved in lipid droplet biogenesis (Ldb16; mRNA found upregulated in RNA seq) was essential for genipin-induced cytoprotection (ldb16 deletion mutants still showed 5% cell death upon Syn expression, with and without addition of genipin), suggesting a link between genipin cytoprotection and lipid droplet biogenesis. Comparing microscopic analysis of Syn-GFP in wild type (presented in Fig. 2B) and in ldb16 mutants (presented in Fig. 6K), it seems that loss of ldb16 in general prominently induces the aggregation of Syn. This

needs to be assessed and directly compared in wt and *ldb16* mutants using same setting. It seems that loss of *ldb16* per se massively induces the aggregation of Syn (e.g numerous inclusions per cell), thus it is a bit unclear what to conclude here.

The results presented in figure 2 were obtained using W303.1. A strain with two copies of human aSyn-GFP stably inserted. This model results in mild phenotypes as described by us and others in several papers^{1,3-6}. Regarding the results on lipid storage presented in figure 6, as we are using a deletion mutant, a different background strain and expression vector was used the BY4741 strain.

With the BY4741 strain either WT or with the *LDB16* deletion when expressing a multicopy vector coding for aSyn-GFP in fact we can observe an exacerbation in inclusions (comparing Supplementary Figure 5 D for the WT with Figure 6I for the *LDB16* deletion). In this case, the phenotype of aSyn-GFP in the WT is aggravated as the toxicity increases at 6h of expression for around 10% (Supplementary Figure 5C) but importantly we continue to observe a reduction in % of cells with inclusions by genipin (Supplementary Figure 5B). This protection is lost in the *LDB16* deletion (Figure 6I right panel) where the % of cells with inclusion is not affected by genipin.

It is also important to notice that aSyn-GFP toxicity is decreased in *ldb16* deletion, comparing with WT background (comparing Figure 6H with Supplementary Figure 5C) to a % of death similar to the obtained with the W303 background where genipin is very effective (Figure 2A). Indicating that the lost of effect of genipin is not due to the alteration in toxicity.

This observation of increased inclusions in the *ldb16* deletion as noted by the reviewer is actually a very interesting topic, since the impact of aSyn on lipid homeostasis and *vice versa* is not well known or described. We have assessed the aggregates area from figure Supplementary Figure 5 D for the WT and Figure 6I for the *LDB16* deletion with and without genipin that is presented in figure attached.

As clearly observed, we do not have alteration in the area of the aggregates in the *LDB16* deletion with and without genipin confirming our conclusion that genipin protection is dependent of *LDB16*.

As included in discussion “Fanning and colleagues described that the prevention of lipid droplet formation, either by genetic manipulation or by a pharmacological approach, could increase aSyn toxicity. Also, they proposed a possible compensatory mechanism where aSyn mediated lipid dis-homeostasis is initially compensated by an increase in lipid storage and therefore protecting cells from aSyn toxicity”. Even though they do not assess aSyn aggregation status, they proved that the content in di- and tri-glycerides strongly impacts on aSyn toxicity.

We included in text this interesting observation of the increase of aggregates between WT and *LDB16* deletion, but since it is out of our scope to uncover fundamental aspects of aSyn pathophysiology we did not include the figure above since we want the manuscript to be focused on genipin effects. Moreover this information to be presented will require not only assessment and direct comparison of area of aggregates in WT and *ldb16* mutants using same setting but also a complete characterization in terms of endocytosis impairment, neutral lipids contents etc. Once more this is an interesting topic for further work on aSyn pathophysiology.

6. In general, the analysis of genipin effects in the *Drosophila* PD model seem a bit limited. While it is of course not necessary to confirm all findings obtained in the yeast system also in the fly model, an expression control for Syn seems necessary. In addition,

it would be beneficial to at least confirm the main effect of genipin (inhibition of Syn aggregation) also in this model system (e.g. via immunoblotting as has been done in the yeast system).

An expression control for aSyn was included, as requested. Importantly, genipin does not induce any changes in aSyn levels in flies' brain, as shown by western blot (new Figure 7C). Moreover, we performed a Triton-X solubility assay and found that genipin promotes a significant reduction of aSyn-GFP in the insoluble fraction (new Figure 7D).

We acknowledge the reviewer's question as the all work was improved by the experiments performed.

In addition, the presented data needs to be improved and necessary controls need to be included in the figure panels:

- The pro-survival effect of genipin on Syn-expressing flies shown in Fig. 7A is very minor. These curves reflect the survival of only 30 flies per genotype, kept in 3 vials of 10 flies each. This needs to be repeated in independent experiments to confirm this effect, as the number of flies is very low and the effect small. In addition, control flies treated with genipin need to be included in the same experiments (in particular as genipin seems to affect control flies in a non-concentration dependent manner according to Supplemental Fig. 6 where some concentrations of genipin seemed to initially kill the control flies, some seemed to slightly rescue). Thus, all four groups need to be compared directly and statistics need be adjusted to compare these four groups.

All the experiments were repeated and the controls were performed and added, as suggested. Importantly, the effect of genipin in the survival rate is much clear now with higher statistical significance.

In general, control flies treated with genipin should be included in the analyses in Figure 7. For Figure 7B, a two-way ANOVA needs to be performed (just as done by the authors for Supplemental Figure 6B, showing corresponding climbing activity in the control flies.) Moreover, control flies treated with the same concentration of genipin should be shown in Fig. 7B.

All the experiments were repeated, as suggested and a new figure 7B included. All four groups were compared directly, and statistics were performed as suggested by TWO-WAY ANOVA using a matching factor where each row represents a time point. Multiple comparisons were performed using main effects on each row.

In respect to the footprint clustering as well as tripod and tetrapod index: it is a bit

unclear how this adds additional value beyond what is already shown in the climbing activity in 7B.

The use of a kinematic assay was two-fold. First, to understand what is the effect of long-term expression of aSyn on coordinated walking. Secondly, if feeding 1mM genipin could reverse some of these deleterious effects. The results obtain answer our first question and support the results obtained. We updated the text to better clarify the goals of this assay.

Is there any reason to use transgenic flies expressing Syn-GFP for survival and climbing activity and flies expressing untagged Syn for this extended analysis of motor defects?

The intend to use untagged aSyn in the kinematic experiments was to obtain additional behavioural data in the most physiological and sensitive way possible. Also, the parameters measured by the FlyWalker are more specific and detailed than just the climbing capacity of the flies. This could allow us to further investigate why and how the climbing activity of the animals are affected, even in a more physiological scenario.

The quantification of the footprint analysis shown in Fig. 7C is not convincing. There is no significant effect between Syn and Syn+genipin. Thus, the conclusion here would be that genipin has no significant rescuing effect.

As previously stated, one of the goals of the FlyWalker assay was to understand the effect of long-term expression of aSyn on coordinated walking, as observed in the climbing assay. In this regard, we present several parameters that answer this question, including footprint and gait analysis. We also updated our script (new figure F) to better quantify the stance phase of the animals and included an additional parameter we termed *Stance straightness*, which measures the “wobbliness” of the stance phase relative to the body axis.

Regarding the footprint clustering parameter, we agree the rescue effect of 1 mM genipin is not sufficiently robust to show statistical significance relative to aSyn expression. Nevertheless, this parameter nicely describes the deleterious effect of aSyn on motor performance in addition to be consistent with the effect visualized by the remaining metrics.

We updated the text in order to better illustrate these concepts.

Minor:

Line 56-58: The following statement should be supported by references:

“However, in pathological conditions, alpha-Syn is described to exert toxic effects, not only by its aggregation process, but also by performing excessive membrane interactions”

The sentence was rewritten to make it clearer and a reference was added as requested

Line 131: Please correct, the sentence is unclear

"..., yeast cells tend to form less detergent-insoluble with SDS or a greater portion of smaller inclusions"

Sentence was re-written to make it more clear

Line 111: The authors do not find any effect of genipin on growth arrest induced by FUS and Htt expression in yeast and thus state "The lack of effect of genipin on cell growth in these models reinforces the potential of genipin to be a specific bioactive molecule to counteract alphaSyn-GFP toxicity and aggregation". As genipin has been shown by others to be neuroprotective in different model systems and scenarios, this statement should be corrected.

Sentence was rewritten to clarify that it is the lack of genipin effect to reduce toxicity and aggregation of FUS and Htt expression in yeast that confer specificity to counteract aSyn-GFP toxicity and aggregation.

Bibliography

1. Macedo, D. *et al.* (Poly)phenols protect from α -synuclein toxicity by reducing oxidative stress and promoting autophagy. *Human Molecular Genetics* **24**, 1717–1732 (2015).
2. Hou, Y. C., Tsai, S. Y., Lai, P. Y., Chen, Y. S. & Chao, P. D. L. Metabolism and pharmacokinetics of genipin and geniposide in rats. *Food and Chemical Toxicology* **46**, 2764–2769 (2008).
3. Outeiro, T. F. Yeast Cells Provide Insight into Alpha-Synuclein Biology and Pathobiology. *Science* **302**, 1772–1775 (2003).
4. Tenreiro, S. *et al.* Phosphorylation modulates clearance of alpha-synuclein inclusions in a yeast model of Parkinson's disease. *PLoS Genet* **10**, e1004302 (2014).
5. Macedo, D. *et al.* (Poly)phenols protect from α -synuclein toxicity by reducing oxidative stress and promoting autophagy. *Human Molecular Genetics* **24**, 1717–1732 (2015).
6. Rosado-Ramos, R. *et al.* Small Molecule Fisetin Modulates Alpha-Synuclein Aggregation. *Molecules* **26**, 3353 (2021).

Reviewer #4 (Remarks to the Author):

The paper focuses on the investigation of a mixture of natural leaf extracts from *Corema album*, previously shown to have protective effects against alpha-synuclein pathology. Genipin is identified as the major responsible for this effect and several complementary techniques are used to demonstrate and understand the origins of this effect. The topic is certainly interesting and it builds on previous work by the PI. The data convincingly show a reduction in the formation of alpha-synuclein aggregates in presence of genipin. The authors report insights achieved through NMR on different parts of the protein in a residue resolved manner. Interestingly different effects are monitored along the primary sequence ranging from intensity changes (N-ter), to essentially no effect (C-ter) to a combination of intensity changes/chemical shift perturbations (60-80, 100-120) residues. It would be interesting to discuss these findings more in detail also in terms of possible structural and dynamic variations induced by genipin and compare them to effects of other compounds that are known to interact with alpha-synuclein (EGCG for example).

Based in NMR data, the interaction residues are clearly lysines. We include this new information in results section: "The reduced signal intensity correlates with the presence of lysines (Figure 3G; blue dots) and can be explained by genipin-mediated cross-linking of lysines. Interestingly, aSyn has no arginines so all fully positively charged residues at physiological pH are lysines making aSyn a good target for genipin. Based on the literature^{7,8}, NMR data could be interpreted as a covalent modifications (cross-linking) of lysines and due to that we highlighted them in Figure 3G.

The type of interaction between genipin and aSyn that our NMR data reveal is quite different than what is described for EGCG with aSyn monomer. This comparison was included and discussed as suggested.

Moreover we performed a preliminary docking analysis considering the conditions described in Himadri Shekhar Baul et al⁹ using the aSyn micelle-bound human a-synuclein structure 1XQ8 and it also reveals some differences in the type of interactions predicted for genipin and EGCG.

Table 1: Molecular docking study of human alpha synuclein with ligands Genipin and EGCG.

Ligand name	Binding energy (Kcal/Mol)	Docking pose	Docking pose close up	2D protein-ligand interactions
Genipin	-3,3			EGCG	-4,1			
Genipin could bind to aSyn by:

Alkyl - LYS34, LYS37

Van der Waals interactions- Ala30, Thr3

EGCG could bind to aSyn by:

Pi-Sigma - ALA30

Alkyl - LYS 34

Van der Waals interactions - Thr 33, Val 37, Gly 31, Val 26, Ala 29

A more detailed and in-depth docking analysis could be performed if needed.

The authors also discuss the link between alpha-synuclein and lipid metabolism (paragraph 7). Would it be possible to access residue resolved information on this topic? For example by investigating by NMR the properties of alpha-synuclein in presence of lipids, or of membrane models, with and without genipin?

Actually, it would be interesting to investigate by NMR the properties of alpha-synuclein in presence of lipids, or of membrane models, with and without genipin as suggested. However for the experiment in presence of lipids, the residues affected by genipin are the residues that form the helices inserted in the liposomes, so the line-

broadening makes them undetectable by solution NMR. Anyway we would expect that liposome interacting aSyn would not be accessible to genipin or at least the interaction would be heavily restricted.

Bibliography

7. Wang, S. S. S., Hsieh, P.-L., Chen, P.-S., Chen, Y.-T. & Jan, J.-S. Genipin-cross-linked poly(L-lysine)-based hydrogels: synthesis, characterization, and drug encapsulation. *Colloids Surf B Biointerfaces* **111**, 423–431 (2013).
8. Mekhail, M., Jahan, K. & Tabrizian, M. Genipin-crosslinked chitosan/poly-L-lysine gels promote fibroblast adhesion and proliferation. *Carbohydrate Polymers* **108**, 91–98 (2014).
9. Baul, H. S. & Rajiniraja, M. Favorable binding of Quercetin to α -Synuclein as potential target in Parkinson disease: An *In silico* approach. *Rese. Jour. of Pharm. and Technol.* **11**, 203 (2018).

Reviewer #5 (Remarks to the Author):

The manuscript is dedicated to the study of anti-aggregation properties of the Genipin substance from *Corema album*. The author have shown that genipin affects the alpha-synuclein aggregation and toxicity. Different plants are actively used as a source of anti-amyloid agents. The author have found not only that the leaves contain some substance, but also identified it, which allowed more detailed study of its effects. The genipin has been shown non-toxic for yeast and fruit flies, and prevent the effects of alfa-synuclein overproduction. Though the results provided in the manuscript are very interesting, there are some comments:

1. The yeast-based model are actively used for study of alfa-synuclein aggregation. Though it is an adequate model for the study, a lot of molecular systems in yeast cells are quite different. The same with fruit flies. Have the author tested the effects of genipin in mammal-based models?

2. The same question with the metabolic changes in yeast cells. The molecular systems might be quite different, so it would be better to show similar results with transcriptomic analysis in mammals.

Answering to question 1 and 2:

Currently, there is a huge lack of models that mimic the process of formation of LB-like inclusions in cells just based on the expression of wild type (WT) aSyn. As stated, the fact that aSyn do not aggregate spontaneously in mammalian cells as in yeast cells makes it very difficult to study in these systems. However, the cell model commonly described by Tiago Outeiro (co-author in the current paper) synuclein-GFP (in a truncated form) is transfected into H4 cells and co-expressed with synphilin-1, a known aSyn-interacting protein that is also present in LBs, to further promote its aggregation¹⁰. This system can be used to answer the question although is also controversial. Moreover, we are aware that the system does not work in SH-SY5Y and LUHMES (Tiago Outeiro, personally communicated). To further add strength to our results we decided to go *in vivo* and used a fly model of aSyn aggregation.

Moreover, the use of yeast cells to modulate central and key molecular mechanisms associated with neurodegenerative diseases has led, so far, to important advances in the field. *Saccaromyces cerevisiae* was used as a model for the study of countless hallmarks of neurodegenerative diseases and, specifically, for the aggregation process so important in these diseases.

Wang et al. (2014) used the yeast model to test a potential mechanism for aSyn aggregation and found that phosphatidylethanolamine (PE) deficiency in yeast cells causes ER stress, vesicle defects, aSyn aggregation, and cells death¹¹. A study by sought to tackle the relationship between aSyn aggregation, lipid homeostasis, and cellular toxicity. The authors performed lipidomic profiling in yeast displaying aSyn aggregation

and found that dysfunctional lipid homeostasis, induced by aSyn aggregation, leads to cytotoxicity due to the accumulation of oleic acid (OA) and diglycerides (DG) in lipid droplets of aSyn-expressing yeast¹². Yeast is not only an excellent model for identifying novel pathophysiological mechanisms but also for probing and testing chemical and genetic modifiers of a disease^{4-6,13-16} conducted a screen for genetic modifiers of aSyn aggregation and identified 33 genes that modulate aSyn aggregation and cytotoxicity¹⁷. One of these modifiers, Pah1, is an enzyme that converts PA to DG in yeast. In accordance with the findings of Fanning, Soste found that decreasing DG levels through the inhibition of Pah1 ameliorates aSyn aggregation and cytotoxicity. This suggests a key role for DG and lipid droplet homeostasis in aSyn aggregation and cytotoxicity in PD¹⁷. Collectively, these studies demonstrate the power of the yeast system for investigating the lipid-related pathophysiology of two prominent neurodegenerative disorders. The unicellular nature of yeast makes it particularly straightforward to track biochemical changes in molecules such as lipids. Importantly, the capacity described of aSyn to control the vesicular dynamics and vesicular recycling in neurons are maintained when the protein is expressed in yeast cells, making it a prime model to study the biology and pathophysiology of aSyn¹⁸.

Also, *Drosophila melanogaster*, has played a valuable role in the field, taking advantage of its genetic technologies and established biological knowledge. *Drosophila* has many advantages. They are relatively easy to maintain and manipulate, and, most importantly, mutant flies are relatively straightforward and fast to obtain due to a short generation time (about 10 days). Also, its genome sequence has been available since 2000¹⁹ and can be easily manipulated. They have a complex nervous system as it possesses dopaminergic system that are involved in the control of complex motor behaviours, such as walking, climbing or flying, constituting, therefore, a suitable model to study changes in motor capacities. The negative geotaxis assay (so called climbing assay), designed to evaluate *Drosophila's* climbing abilities remains one of the most commonly used, although more complex tasks have been developed^{20,21}. Importantly, *Drosophila* does not have a SNCA ortholog but there are countless models overexpressing human WT or mutant aSyn. These models take advantage of the Gal4/UAS system that allows promoter-dependent cell- or tissue-specific expression of aSyn. The use of the elav-Gal4/UAS system to direct the pan-neural expression aSyn protein in *Drosophila*, induces the formation of aSyn inclusions. aSyn transgenic *Drosophila* models recapitulate the essential features of PD patients as adult-onset loss of DA neurons, intracellular inclusions containing aSyn and locomotor dysfunction. Also, aSyn phosphorylation plays a key role in disease pathogenesis. Ser129 is found extensively phosphorylated in brain tissues from PD patients, suggesting a role for Ser129 phosphorylation in disease pathogenesis. In PD transgenic fly models, it has been shown that human aSyn is phosphorylated at Ser129, and that this phosphorylation increases with age, as well as dopaminergic neurons' degeneration mimicking the pathogenic phenomena in PD patient. PD *Drosophila* models were also used for several pharmacological interventions to modulate aSyn toxicity. As an example, several molecules, as nicotinamide, pergolide, bromocriptine, and 2,3,4,5-tetrahydro-7,8-

dihydroxy-1-phenyl-1H-3-benzazepine (SK&F 38393), D-519, and D-520 were tested for its capacity to modulate aSyn pathology and were found to be effective in improving the locomotor function of PD flies. Several more molecules were already tested in PD fly model too investigate its efficacy and mechanism of action, as reviewed²².

Overall both systems, yeast and flies, have been used to study the pathophysiology of PD and even the effects of small molecules with validated findings in mammalian systems. The fact that mammalian systems of aSyn aggregation are very artificial and without consistent phenotypes lead us to opt for the aforementioned models for our study.

3. Though the authors have precisely described the effects of genipin, no molecular mechanisms explaining the effects are provided. Which proteins does genipin interact with? How does it affect their activity?

As many phytochemicals, genipin is described to have several bioactivities, as stated in the discussion (anti-depressant, neuroprotective, anti-inflammatory, anti-obesity). As well, genipin was shown, within the scope of this paper, to have a pleiotropic effect in yeast cells. As verified by the analysis of RNA-seq, genipin is proposed to act in cell central carbon metabolism, endocytosis and trafficking, as well as in lipid storage and homeostasis. Therefore, we hypothesize that genipin might have several targets within yeast cells and thus, proving genipin-protein interactions would not be possible. However, the most striking observation is the fact that genipin interferes with aSyn aggregation and it was demonstrated that genipin directly interacts with monomeric aSyn (Figure 3) impairing its aggregation which is described as a central pathological hallmark of synucleinopathies. The drosophila experiments reinforces the activity and potential of genipin to reduce death due to asyn expression in vivo (Figure 7A). We highlight this aspect in the discussion and final conclusion.

4. The bioinformatic analysis should be described in more details. The important steps are missing.

The bioinformatic analysis details were added, as suggested.

5. There are no data how the synuclein and genipin affects the neural tissue of flies, only results of physiological tests are provided.

The experiments assessing the aggregation state of aSyn in fly neuronal tissues were performed as suggested and included in figure 7 and discussion.

6. If genipin directly prevents alfa-Syn aggregation by interaction with monomers, it would be better to confirm this interaction by providing its model and mutants unable

to interact with genipin.

Indeed this is a very interesting suggestion. If we consider the NMR data, the interaction residues are clearly lysines. In fact we include this analysis in results section: "The reduced signal intensity correlates with the presence of lysines (Figure 3G; blue dots) and can be explained by genipin-mediated cross-linking of lysines. Interestingly, aSyn has no arginines so all fully positively charged residues at physiological pH are lysines making aSyn a good target for genipin." We may then suppose that any mutation that removes a lysine would affect the interaction, but it will be not very strongly, because genipin wouldn't need so many lysines to produce an effect and therefore the use of mutated forms could be inconclusive.

Moreover the fact that lysines seem to be a good target was explained in the discussion based in the literature: "Moreover, the ability of small molecules to interact with mutant A53T aSyn lysins was already reported. Despite the fact that the interaction was only reported for lysine 23 and 33, the interaction resulted in a decreased aSyn species' toxicity".

Considering the model of interaction genipin-aSyn we have performed a preliminary docking analysis considering the conditions described in Himadri Shekhaer Baul et al.⁹ using the aSyn micelle-bound human a-synuclein structure 1XQ8 and compared the type of interactions predicted for genipin with EGCG, a known aSyn aggregation inhibitor (please see answer to reviewer 4). Although the binding energy is not very low (-3.3 Kcal/Mol) we can perform a more detailed and in-depth docking analysis and included if requested.

References

1. Macedo, D. *et al.* (Poly)phenols protect from α -synuclein toxicity by reducing oxidative stress and promoting autophagy. *Human Molecular Genetics* **24**, 1717–1732 (2015).
2. Hou, Y. C., Tsai, S. Y., Lai, P. Y., Chen, Y. S. & Chao, P. D. L. Metabolism and pharmacokinetics of genipin and geniposide in rats. *Food and Chemical Toxicology* **46**, 2764–2769 (2008).
3. Outeiro, T. F. Yeast Cells Provide Insight into Alpha-Synuclein Biology and Pathobiology. *Science* **302**, 1772–1775 (2003).
4. Tenreiro, S. *et al.* Phosphorylation modulates clearance of alpha-synuclein inclusions in a yeast model of Parkinson's disease. *PLoS Genet* **10**, e1004302 (2014).
5. Macedo, D. *et al.* (Poly)phenols protect from α -synuclein toxicity by reducing oxidative stress and promoting autophagy. *Human Molecular Genetics* **24**, 1717–1732 (2015).
6. Rosado-Ramos, R. *et al.* Small Molecule Fisetin Modulates Alpha-Synuclein Aggregation. *Molecules* **26**, 3353 (2021).

7. Wang, S. S. S., Hsieh, P.-L., Chen, P.-S., Chen, Y.-T. & Jan, J.-S. Genipin-cross-linked poly(L-lysine)-based hydrogels: synthesis, characterization, and drug encapsulation. *Colloids Surf B Biointerfaces* **111**, 423–431 (2013).
8. Mekhail, M., Jahan, K. & Tabrizian, M. Genipin-crosslinked chitosan/poly-L-lysine gels promote fibroblast adhesion and proliferation. *Carbohydrate Polymers* **108**, 91–98 (2014).
9. Baul, H. S. & Rajiniraja, M. Favorable binding of Quercetin to α -Synuclein as potential target in Parkinson disease: An *In silico* approach. *Rese. Jour. of Pharm. and Technol.* **11**, 203 (2018).
10. Masaracchia, C. *et al.* Molecular characterization of an aggregation-prone variant of alpha-synuclein used to model synucleinopathies. *Biochimica et Biophysica Acta (BBA) - Proteins and Proteomics* **1868**, 140298 (2020).
11. Wang, S. *et al.* Phosphatidylethanolamine deficiency disrupts α -synuclein homeostasis in yeast and worm models of Parkinson disease. *Proc. Natl. Acad. Sci. U.S.A.* **111**, (2014).
12. Fanning, S. *et al.* Lipidomic Analysis of α -Synuclein Neurotoxicity Identifies Stearoyl CoA Desaturase as a Target for Parkinson Treatment. *Molecular Cell* **73**, 1001-1014.e8 (2019).
13. Griffioen, G. *et al.* A yeast-based model of alpha-synucleinopathy identifies compounds with therapeutic potential. *Biochim Biophys Acta* **1762**, 312–318 (2006).
14. Williams, R. B. *et al.* Bioactivity profiling with parallel mass spectrometry reveals an assemblage of green tea metabolites affording protection against human huntingtin and α -synuclein toxicity. *Journal of Agricultural and Food Chemistry* **55**, 9450–9456 (2007).
15. Su, L. J. *et al.* Compounds from an unbiased chemical screen reverse both ER-to-Golgi trafficking defects and mitochondrial dysfunction in Parkinson's disease models. *Disease Models & Mechanisms* **3**, 194–208 (2010).
16. Macedo, D. *et al.* (Poly)phenol-digested metabolites modulate alpha-synuclein toxicity by regulating proteostasis. *Sci Rep* **8**, 6965 (2018).
17. Soste, M. *et al.* Proteomics-Based Monitoring of Pathway Activity Reveals that Blocking Diacylglycerol Biosynthesis Rescues from Alpha-Synuclein Toxicity. *Cell Systems* **9**, 309-320.e8 (2019).
18. Zabrocki, P. *et al.* Phosphorylation, lipid raft interaction and traffic of α -synuclein in a yeast model for Parkinson. *Biochimica et Biophysica Acta (BBA) - Molecular Cell Research* **1783**, 1767–1780 (2008).
19. Myers, E. W. *et al.* A Whole-Genome Assembly of *Drosophila*. *Science* **287**, 2196–2204 (2000).
20. Sang, T.-K. *et al.* A *Drosophila* Model of Mutant Human Parkin-Induced Toxicity Demonstrates Selective Loss of Dopaminergic Neurons and Dependence on Cellular Dopamine. *Journal of Neuroscience* **27**, 981–992 (2007).
21. Ali, Y. O., Escala, W., Ruan, K. & Zhai, R. G. Assaying Locomotor, Learning, and Memory Deficits in *Drosophila* Models of Neurodegeneration. *JoVE* 2504 (2011) doi:10.3791/2504.
22. Xiong, Y. & Yu, J. Modeling Parkinson's Disease in *Drosophila*: What Have We Learned for Dominant Traits? *Front. Neurol.* **9**, 228 (2018).

REVIEWER COMMENTS

Reviewer #1 (Remarks to the Author):

The authors addressed all my comments on the HPLC and mass spectrometry part adequately. Therefore, I suggest that publication is justified.

However, during revision of the manuscript, some typos were found and should be corrected:

- line 627: please change to "10 μm " (not "10 mm")
- line 633: it should read "1 column volume"
- line 645: "0.45- μm filter" instead of "0.45-mm filter"
- line 651: column description: please specify the column length (150 mm), the inner diameter (3 or 2 mm?), and the particle diameter, e.g. 3 μm
- line 676: "13C" instead of "C13" (with "13" in superscript)

Reviewer #3 (Remarks to the Author):

In this revised version of the manuscript, Rosado-Ramos et al. now provide additional data that clarify some of my previous concerns. In particular the *Drosophila* survival analyses and the climbing activity have been substantially improved. Moreover, the additional analysis of Syn aggregation in *Drosophila* brain lysates nicely complements the initial data that genipin prevents Syn aggregation in vitro as well as in the yeast model.

It still remains unclear whether any of the hits identified to be up/downregulated upon genipin treatment (transcriptomics) is involved in mediating cytoprotection. The respective results section is very descriptive, lists the different changes in transcript levels and metabolites without reaching clear conclusions as to what this could imply and without testing causality of any of these effects. Thus, in my view, the title is still overstating the actual results. More particularly, the data connecting lipid metabolism, Ldb16 and Syn toxicity remains premature (as detailed below). However, though the authors were not able to provide more details concerning the mode of action of genipin, potential mechanisms of cytoprotection are now discussed in new sections of the discussion and the finding that genipin counteracts Syn aggregation is important.

Main:

In respect to my previous concern regarding the data presented for the ldb16 mutant: the authors now argue that this cannot be directly compared with the data shown in Figure 1, as another wild type strain (W303 versus BY4741) was used. Thus, it becomes even more important to include data on the corresponding wild type for the ldb16 mutant to support the conclusions. The authors now present the corresponding wild type, but only in Supplemental Fig. 5, which complicates any direct comparison with the ldb16 mutant (in Fig. 6) for the reader, as one has to switch back and forth to appreciate any differences. Thus, the corresponding wild type for the ldb16 mutant is required directly in the respective panels showing the effect on toxicity (PI staining, 6i) and on Syn aggregation (microscopy, now Fig. 6j, k).

The authors argue that in the BY4741 background, Syn is more toxic than in the W303 background (which, if I understand correctly, has been used for all previous Figures). How strong is the genipin protection in the BY4741 background? Judging from Supplemental Figure 5b, the rescuing effect of genipin indeed seems absent in the BY4741 strain (or minor, no significances are indicated in the respective panel).

To conclude that the presence of Ldb16 is essential for genipin cytoprotection, it is not only critical to show the ldb16 mutant and the corresponding wild type (BY4741) in the same panel (for toxicity as well as aggregation), but also appropriate statistics need to be done for the comparison wt and ldb16 treated or not with genipin.

Similarly, also the new data that have been included in the erg6 mutant are in the BY4741 background. Also here, the corresponding wild type control has to be included in every panel

to allow any conclusion.

Minor:

Indicators of significance might have been lost in Fig. 6g and 6h, as the lines indicating the different comparisons are shown, but asterisks are missing.

The text might benefit from careful proof reading, as some sentences (in particular new parts in the discussion) seem to lack words and contain errors.

Reviewer #5 (Remarks to the Author):

I would like to thank the authors for the great improvement of the text and for the answering my comments. Several comments:

1. As far as I understand, the effects of genipin on the aggregation of alpha-Syn is not explained only by direct interaction between genipin and alpha-Syn. So, the genipin also alters some cellular processes, which affect synuclein aggregation, doesn't it? This question should be clarified in the conclusion.
2. Regarding my previous 1 and 2 questions, I mentioned the effects of genipin on the yeast cells, not on the aggregation of alpha-Syn. If genipin affects some process apart from the synuclein aggregation, the effect might be species specific. That is why I suggested to do RNA-Seq with genipin in eukaryotic cells.
3. I suggest restructuring the discussion. The enumeration of different processes is difficult to follow and is a bit confusing.

Minor comments:

1. It is not clear why the authors concluded that genipin interacts with monomeric alpha-Syn. It might interact with other forms including intermediate oligomeric.
2. I could not find how many replicates have been made for RNA-Seq.
3. I could not find whether the raw result of RNASeq is publicly available.

Reviewer #1 (Remarks to the Author):

The authors addressed all my comments on the HPLC and mass spectrometry part adequately. Therefore, I suggest that publication is justified.

However, during revision of the manuscript, some typos were found and should be corrected:

- line 627: please change to "10 µm" (not "10 mm")
- line 633: it should read "1 column volume"
- line 645: "0.45-µm filter" instead of "0.45-mm filter"
- line 651L: column description: please specify the column length (150 mm), the inner diameter (3 or 2 mm?), and the particle diameter, e.g. 3 µm
- line 676: "13C" instead of "C13" (with "13" in superscript)

All the typos were corrected and column details were added.

Reviewer #3 (Remarks to the Author):

In this revised version of the manuscript, Rosado-Ramos et al. now provide additional data that clarify some of my previous concerns. In particular the *Drosophila* survival analyses and the climbing activity have been substantially improved. Moreover, the additional analysis of Syn aggregation in *Drosophila* brain lysates nicely complements the initial data that genipin prevents Syn aggregation in vitro as well as in the yeast model.

It still remains unclear whether any of the hits identified to be up/downregulated upon genipin treatment (transcriptomics) is involved in mediating cytoprotection. The respective results section is very descriptive, lists the different changes in transcript levels and metabolites without reaching clear conclusions as to what this could imply and without testing causality of any of these effects. Thus, in my view, the title is still overstating the actual results. More particularly, the data connecting lipid metabolism, Ldb16 and Syn toxicity remains premature (as detailed below). However, though the authors were not able to provide more details concerning the mode of action of genipin, potential mechanisms of cytoprotection are now discussed in new sections of the discussion and the finding that genipin counteracts Syn aggregation is important.

Thank you for your comment. Indeed, giving the pleiotropic effect of the small molecule genipin, a single and specific target could not be claimed based in our

data. Our hypothesis is that genipin can protect against α Syn deleterious effects acting on several cellular processes at the same time, as well as directly on α Syn. That is why, there are some effects that are only descriptive and cannot be claimed as causative or not. Nevertheless, we revised the title and improved the discussion to better reflect our point of view on this topic.

Main:

In respect to my previous concern regarding the data presented for the *ldb16* mutant: the authors now argue that this cannot be directly compared with the data shown in Figure 1, as another wild type strain (W303 versus BY4741) was used. Thus, it becomes even more important to include data on the corresponding wild type for the *ldb16* mutant to support the conclusions. The authors now present the corresponding wild type, but only in Supplemental Fig. 5, which complicates any direct comparison with the *ldb16* mutant (in Fig. 6) for the reader, as one has to switch back and forth to appreciate any differences. Thus, the corresponding wild type for the *ldb16* mutant is required directly in the respective panels showing the effect on toxicity (PI staining, 6i) and on Syn aggregation (microscopy, now Fig. 6j, k). The authors argue that in the BY4741 background, Syn is more toxic than in the W303 background (which, if I understand correctly, has been used for all previous Figures). How strong is the genipin protection in the BY4741 background? Judging from Supplemental Figure 5b, the rescuing effect of genipin indeed seems absent in the BY4741 strain (or minor, no significances are indicated in the respective panel). To conclude that the presence of *Ldb16* is essential for genipin cytoprotection, it is not only critical to show the *ldb16* mutant and the corresponding wild type (BY4741) in the same panel (for toxicity as well as aggregation), but also appropriate statistics need to be done for the comparison wt and *ldb16* treated or not with genipin.

The BY4741 background strain was used to compare *ldb16* data (previously on Supp. Figure 5). In fact, as we are using a multi copy vector in this background we have higher toxicity of α Syn. Importantly, genipin retains its protective effect statistically significant either for toxicity or for the reduction of aggregation (fig 6l and 6m). We agree that it would be easier for the reader to have this results in the same figure as *ldb16* mutant results, so we added it to the Figure 6, as suggested. The results were updated accordingly.

Similarly, also the new data that have been included in the *erg6* mutant are in the

BY4741 background. Also here, the corresponding wild type control has to be included in every panel to allow any conclusion.

We agree that it would be easier for the reader to have BY4741 background results in the same figure as the *erg6* mutant results, so we added it to the Figure 6, as suggested. The results were updated accordingly.

Minor:

Indicators of significance might have been lost in Fig. 6g and 6h, as the lines indicating the different comparisons are shown, but asterisks are missing.

Figure 6g and Figure 6H were corrected statistical significance.

The text might benefit from careful proof reading, as some sentences (in particular new parts in the discussion) seem to lack words and contain errors.

All text was carefully revised and errors corrected

Reviewer #5 (Remarks to the Author):

I would like to thank the authors for the great improvement of the text and for the answering my comments. Several comments:

1. As far as I understand, the effects of genipin on the aggregation of alpha-Syn is not explained only by direct interaction between genipin and alpha-Syn. So, the genipin also alters some cellular processes, which affect synuclein aggregation, doesn't it? This question should be clarified in the conclusion.

Indeed, the action of genipin directly on the structure of α Syn seems to be one of the events that ultimately leads to cellular protection. And, in fact, genipin interferes in several central cellular processes affected in α Syn pathophysiology. This question was clarified in discussion and the final conclusion as suggested.

2. Regarding my previous 1 and 2 questions, I mentioned the effects of genipin on the yeast cells, not on the aggregation of alpha-Syn. If genipin affects some process apart from the synuclein aggregation, the effect might be species specific. That is why I suggested to do RNA-Seq with genipin in eukaryotic cells.

As discussed, we believe that genipin interferes with other cellular processes affected in the α Syn pathophysiology. The fact that our transcriptomic analysis only reveals genes affected by genipin under α Syn expression lead us to follow the hypothesis that genipin protective action is dependent of α Syn aggregation- and toxicity-induced impairments. This was the reason to select yeast and flies models. We intended to study the effect of genipin on the pathophysiology of PD avoiding mammalian systems of α Syn aggregation that are very artificial and lacking consistent phenotypes. The following steps should indeed proceed to pre-clinical models using mammalian cells for testing genipin effects besides α Syn aggregation like cell toxicity and ADME properties. In this study, we consider important to first focus in establish how genipin acts in the α Syn-induced impaired cellular processes. This concept was reinforced in the final conclusion.

3. I suggest restructuring the discussion. The enumeration of different processes is difficult to follow and is a bit confusing.

Discussion was revised and re-organized to clarify the different processes as suggested.

Minor comments:

1. It is not clear why the authors concluded that genipin interacts with monomeric alpha-Syn. It might interact with other forms including intermediate oligomeric.

Indeed, it would be possible, and it is very likely, that genipin interacts with other forms of α Syn. However, the NMR signal that we are measuring is only the signal of monomeric α Syn, that is why we can only claim that genipin interacts with the monomer. It is exactly because of this that we suggest in the end of the discussion of the structural alterations that further structural studies should investigate whether genipin might also be able to disassemble preformed, β -sheet-rich structures as well as early intermediates of fibrillogenesis through the same type of interactions. We revise to text to clarify this topic.

2. I could not find how many replicates have been made for RNA-Seq.

The information was added.

3. I could not find whether the raw result of RNASeq is publicly available.

The data will be publicly available at the time of publication.